

**Blowing Snow Sublimation and Transport over Antarctica from 11 Years of CALIPSO**
**Observations**
Stephen P Palm[1], Vinay Kayetha[1], Yuekui Yang[2] and Rebecca Pauly[1]
[1]Science Systems Applications Inc., 10210 Greenbelt Road, Greenbelt, Maryland USA 20771.
[2]NASA Goddard Space Flight Center, Greenbelt, Maryland USA 20771.
Address for all correspondence:
Stephen Palm, Code 612, NASA Goddard Space Flight Center, Greenbelt, Maryland USA 20771.
Email: stephen.p.palm@nasa.gov
Phone: +1-301-614-6276
**ABSTRACT**
Blowing snow processes commonly occur over the earth's ice sheets when near surface wind
speed exceeds a threshold value. These processes play a key role in the sublimation and re-
distribution of snow thereby influencing the surface mass balance. Prior field studies and
modeling results have shown the importance of blowing snow sublimation and transport on the
surface mass budget and hydrological cycle of high latitude regions. For the first time, we
present continent-wide estimates of blowing snow sublimation and transport over Antarctica
based on direct observation of blowing snow events. We use an improved version of the
blowing snow detection algorithm developed for previous work that uses atmospheric
backscatter measurements obtained from the CALIOP lidar aboard the CALIPSO satellite. The
blowing snow events identified by CALIPSO and meteorological fields from MERRA-2 are used
to compute the sublimation and transport rates. Our results show that maximum sublimation
occurs along and slightly inland of the coastline. This is contrary to the observed maximum
blowing snow frequency which occurs over the interior. The associated temperature and
moisture re-analysis fields likely contribute to the spatial distribution of the maximum
sublimation values. However, the spatial pattern of the sublimation rate over Antarctica is
consistent with modeling studies and precipitation estimates. Overall, our results show that
Antarctica average integrated blowing snow sublimation is about 393.4 ± 138 Gt yr$^{-1}$ which is



considerably larger than previous model-derived estimates. We find maximum blowing snow
transport amount of 5 Megatons km$^{-1}$ yr$^{-1}$ over parts of East Antarctica and estimate that the
average snow transport from continent to ocean is about 3.68 Gt yr$^{-1}$. These continent-wide
estimates are the first of their kind and can be used to help model and constrain the surface-
mass budget over Antarctica.
**Keywords:** Blowing snow, sublimation, transport, CALIPSO, Antarctica, surface mass balance

**1 Introduction**
The surface mass balance of the earth's great ice sheets that cover Antarctica and Greenland is
one of today's most important topics in climate science. The processes that contribute to the
mass balance of a snow or ice-covered surface are precipitation, surface runoff (melting),
removal/loss (erosion/sublimation) and deposition of snow mass (wind transport). Sublimation
of snow can occur at the surface but is greatly enhanced within the atmospheric column of the
blowing snow layer. The contributions of these processes to the mass balance vary greatly
spatially, and can be highly localized and very difficult to quantify.
It is well known that the Arctic is experiencing rapid warming and loss of sea ice cover and
thickness. In the past few decades, the Arctic has seen an increase in average surface air
temperature by 2 °C (Przybylak, 2007). Modeling studies suggests an increase in annual mean
temperatures over the Arctic by $8.5 \pm 4.1$ °C over the current century that could lead to a
decrease in sea ice cover by $49 \pm 18$ % (Bintanja and Krikken, 2016). While the Antarctic has
experienced an increase in average surface temperature, most of the warming is observed over
West Antarctica at a rate of 0.17 °C per decade from 1957 to 2006 (Steig et al., 2009; Bromwich
et al., 2013). Such surface warming undoubtedly has implications for ice sheet mass balance
and sea level rise mainly through the melting term of the mass balance equation. However, the
other processes affecting the mass balance of ice sheets may also be experiencing changes that
are difficult to identify and quantify. For instance, models have shown that in a warming
climate, precipitation should increase over Antarctica and most of it will fall as snow (Church et



al., 2013). If snowfall is increasing, perhaps the frequency of blowing snow and subsequently the magnitude of transport and sublimation will increase as well. Thus, understanding how these processes affect the overall mass balance of the ice sheets and how they may be responding to a changing climate, is of growing concern.

In addition to ice sheet mass balance, sublimation of blowing snow is also important for the atmospheric moisture budget in high latitudes. For instance, in the Canadian Prairies and parts of Alaska sublimation of blowing snow was shown to be equal to 30 % of annual snowfall (Pomeroy et al., 1997). About 50 % of the wind-transported snow sublimates in the high plains of southeastern Wyoming (Tabler et al., 1990). Adequate model representation of sublimation processes are important to obtain reliable prediction of spring runoff and determine the spatial distribution/variability of energy and water fluxes and their subsequent influence on atmospheric circulation in high latitude regions (Bowling et al., 2004).

Over Antarctica, blowing snow occurs more frequently than anywhere else on earth. Models driven by long-term surface observations over the Neumayer station (East Antarctica), estimate that blowing snow sublimation removes up to 19 % of the solid precipitation (Van den Broeke et al., 2010). Over certain parts of the Antarctica, where persistent katabatic winds prevail, blowing snow sublimation is found to remove up to 85 % of the solid precipitation (Frezzotti et al., 2002). Over coastal areas up to 35% of the precipitation may be removed by wind through transport and sublimation (Bromwich 1988). Das et al., (2013) concluded that ~ 2.7–6.6 % of the surface area of Antarctica has persistent negative net accumulation due to wind scour (erosion and sublimation of snow). These studies show the potential role of the blowing snow sublimation process in the surface mass balance of the earth's ice sheets.

For the current work, we focus on blowing snow processes over the Antarctic region. Due to the uninhabited expanse of Antarctica and the lack of observations, prior, continent-wide studies of blowing snow sublimation over Antarctica had to rely on parameterized methods that use model re-analysis of wind speed and low level moisture. The presence of blowing snow is inferred from surface temperature, wind speed and snow age (if known). In a series of papers on the modeling of blowing snow, Dery and Yau (1998, 1999, 2001) develop and test a parameterization of blowing snow sublimation. Their results show that most sublimation occurs



along the coasts and over sea ice with maximums in some coastal areas of 150 mm snow water
equivalent (swe). Lenearts et al., (2012a) utilized a high resolution regional climate model
(RACMO2) to simulate the surface mass balance of the Antarctic ice sheet. They found drifting
and blowing snow sublimation to be the most significant ablation term reaching values as high
as 200 mm $yr^{-1}$ swe along the coast. There has been some work done on blowing snow
sublimation from field measurements, but the data are sparse and measurements are only
available within the surface layer (< 10 m). For instance, average monthly rates of sublimation
calculated for Halley Station, Antarctica, varied between 0.13 to 0.44 mm day−1 (King et al.

95    2001).

While transport of blowing snow is considered to be less important than sublimation in terms
of mass balance of the Antarctic ice sheet, erosion and transport of snow by wind can be
considerable in certain regions. Das et al., (2013) have shown that blue ice areas are frequently
seen in Antarctica. These regions exhibit a negative mass balance as all precipitation that falls is
either blown off or sublimated away. Along the coastal regions it has been argued that
considerable mass is transported off the coast via blowing snow in preferential areas dictated
by topography (Scarchilli et al., 2010). In the Tera Nova Bay region of East Antarctica, manned
surface observations show that drifting and blowing snow occurred 80 % of the time in fall and
winter and cumulative snow transport was 4 orders about of magnitude higher than snow
precipitation. Much of this airborne snow is transported off the continent producing areas of
blue ice. Such observations raise questions as to how often and to what magnitude continent to
ocean transport occurs. This is important, particularly for Antarctica where the coastline
stretches over 20,000 km in length and where prevailing strong winds through most of the year.
Due to the sparsity of observations, the only way to estimate the mass of snow being blown off
the coast of Antarctica is by using model parameterizations. Now, for the first time, satellite
observations of blowing snow can help better ascertain the magnitude of this elusive quantity.
Considering that the accuracy of model data is questionable over Antarctica, and the
complicated factors that govern the onset of blowing snow, it is difficult to assess the accuracy
of the parameterization of blowing snow sublimation and transport. Recently, methods have



been developed to detect the occurrence of blowing snow from direct satellite observations. Palm et al., (2011) show that blowing snow is widespread over much of Antarctica and, in all but the summer months, occurs over 50 % of the time over large areas of East Antarctica. In this paper, we present a technique that uses direct measurements of blowing snow from the CALIPSO satellite lidar combined with The Modern-Era Retrospective analysis for Research and Applications, Version 2 (MERRA-2) re-analysis fields of moisture, temperature and wind to quantify the magnitude of sublimation and mass transport occurring over most of Antarctica (north of 82 south). Section 2 discusses the method used to compute blowing snow sublimation from CALIPSO and MERRA-2 data. In Sect. 3 we show results and compare with previous estimates of sublimation. In Sect. 4 we examine sources of error and their approximate magnitudes. Summary and discussion follow in Sect. 5.

## 2 Method

The method developed for detection of blowing snow using satellite lidar data (both ICESat and CALIPSO) was presented in Palm et al., (2011). That work showed examples of blowing snow layers as seen by the calibrated, attenuated backscatter data measured by the CALIOP instrument on the CALIPSO satellite. CALIOP (Cloud-Aerosol Lidar with Orthogonal Polarization) is a two wavelength (532 and 1064 nm) backscatter lidar with depolarization at 532 nm and has been operating continuously since June of 2006 (Winker et. al., 2009). In the lower 5 km of the atmosphere, the vertical resolution of the CALIOP backscatter profile is 30 m. The CALIOP backscatter profiles are produced at 20 Hz, which is about a horizontal resolution of 330 m along track. The relatively strong backscattering produced by the earth's surface is used to identify the ground bin in each profile. After the ground signal is detected, each 20 Hz profile is examined for an elevated backscatter signal (above a pre-defined threshold) in the first bin above the ground. If found and the surface wind speed is greater than 4 m s$^{-1}$, successive bins above that are searched for a 80 % decrease in signal value, which is then the top of the layer. Limited by the vertical resolution of the signal, our approach has the ability to identify blowing snow layers that are roughly 15-20 m or more in thickness. Thus, drifting snow which is





confined to 10 m or less and occurs frequently over Antarctica would not be reliably detected.
The signal from these layers is likely inseparable from the strong ground return. More
information on the blowing snow detection algorithm can be found in Palm et al., (2011).
For the work done in this paper we have created a new version of the blowing snow detection
algorithm which strives to reduce the occurrence of false positive blowing snow detections. This
is done by looking at both the layer average depolarization ratio and color ratio and limiting the
top height of the layer to 500 m. If a layer is detected, but the top of the layer is above 500 m, it
is not included as blowing snow. This height limit helped screen out diamond dust which often
stretches for a few kilometers vertically and frequently reaches the ground. It was found that
for most blowing snow layers, the depolarization and color ratio (1064/532) averaged about 0.4
and 1.3, respectively (see Fig. 1). If the layer average color or depolarization ratios were out of
pre-defined threshold limits, the layer was rejected. The layer average color ratio had to be
greater than 1.0 and the depolarization ratio greater than 0.25. The large color ratio is
consistent with model simulations for spherical ice particles (Bi et al., 2009). Further, logic was
included to reduce misidentification of low cloud as blowing snow by limiting both the
magnitude and height of the maximum backscatter signal in the layer. If the maximum signal
were greater than $2.0 \times 10^{-1}$ km$^{-1}$ sr$^{-1}$, the layer was assumed cloud and not blowing snow. In
addition, if the maximum backscatter, regardless of its value, occurs above 300 m, the layer is
rejected. These changes to the blowing snow detection algorithm slightly decreased (few
percent) the overall frequency of blowing snow detections, but we believe we have reduced the
occurrence of false positives and the resulting retrievals are now more accurate.
Typically, the blowing snow layers are 100–200 m thick, but can range from the minimum
detectable height (20 - 30 m) to over 400 m in depth (Mahesh et al., 2003). Often they are seen
to be associated with blowing snow storms that cover vast areas of Antarctica and can persist
for days. Blowing snow can occur as frequently as 50 % of the time over large regions of East
Antarctica in all months but December–February and as frequently as 75 % April through
October (Palm et al., 2011). An example of a typical blowing snow layer as seen from the
CALIOP backscatter data is shown in Fig. 1.



## 2.1 Sublimation

Sublimation of snow occurs at the surface but is greatly enhanced when the snow becomes airborne by the action of wind and turbulence. Once snow particles become airborne, their total surface area is exposed to the air. If the relative humidity of the ambient air is less than 100 %, then sublimation will occur. The amount of sublimation is dictated by the number of snow particles in suspension and the relative humidity and temperature of the air. Thus, to estimate sublimation of blowing snow, we must be able to derive an estimate of the number density of blowing snow particles and have knowledge of atmospheric temperature and moisture within the blowing snow layer. The only source of the latter, continent wide at least, is from global or regional models or re-analysis fields. The number density of blowing snow particles can be estimated directly from the CALIOP calibrated, attenuated backscatter data if we can estimate the extinction within the blowing snow layer and have a rough idea of the blowing snow particle radius. The extinction can be estimated from the backscatter through an assumed extinction to backscatter ratio (lidar ratio) for the layer. The lidar ratio, though unknown, would theoretically be similar to that of cirrus clouds, which has been extensively studied. Work done by Josset et al., (2012) and Chen et al., (2002) shows that the extinction to backscatter ratio for cirrus clouds typically ranges between 25 and 30 with an average value of 29. However, the ice particles that make up blowing snow are more rounded than the ice particles that comprise cirrus clouds and are on average somewhat smaller Walden et al., (2003). For this paper, we use a value of 25 for the extinction to backscatter ratio.

Measurements of blowing snow particle size have been made by a number of investigators [Schmidt, 1982; Mann et al., 2000; Nishimura and Nemoto, 2005; Walden et al. 2003; Lawson et al., 2006; Gordon and Taylor, 2009], but they were generally made within the first few meters of the surface and may not be applicable to blowing snow layers as deep as those studied here. Most observations have shown a height dependence of particle size ranging from 100 to 200 µm in the lower tens of centimeters above the surface to 50–60 µm near 10 m height (Nishimura and Nemoto, 2005). A notable exception is the result of Harder et al., (1996) at the South Pole, who measured the size of blowing snow particles during a blizzard by collecting



them on a microscope slide. They report nearly spherical particles with an average effective
radius of 15 μm, but the height at which the measurements were made is not reported. From
surface observations made at the South Pole, Walden et al., (2003) and Lawson et al., (2006)
report an average effective radius for blowing snow particles of 19 and 17 μm, respectively.
While no field-measured values for particle radii above roughly 10 m height are available,
modeling work indicates that they approach an asymptotic value of about 10-20 μm at heights
of 200 m or more (Dery and Yau, 1998). It is also reasonable to assume that snow particles that
are high up in the layer are smaller since they have spent more time aloft and have had a
greater time to sublimate. Based on the available data, we have defined particle radius ($r(z)$,
μm) as a linear function of height:
$r(z) = 40 - \frac{z}{20}$ (1)
Thus, for the lowest level of CALIPSO retrieved backscatter (taken to be 15 m – the center of
the first bin above the surface), r(15) = 39.25 μm and at the highest level (500 m), r(500) = 15
μm.
The blowing snow particle number density $N(z)$ (particles per cubic meter) is:
$N(z) = \frac{(\beta(z) - \beta_m(z))\, S}{2\pi\, r^2(z)}$ (2)
Where $\beta(z)$ is the CALIPSO measured attenuated calibrated backscatter at height $z$ (30 m
resolution), $\beta_m(z)$ is the molecular backscatter at height $z$ and $S$ is the extinction to backscatter
ratio (25). Here $\beta(z)$ represents the atmospheric backscatter profile through the blowing snow
layer. Both $\beta_m(z)$ and $\beta(z)$ have units of m$^{-1}$ sr$^{-1}$. We found that the values of N(z) obtained
from Eq. (2) for the typical blowing snow layer range from about $5.0 \times 10^4$ to $1.0 \times 10^6$ particles
per cubic meter. This is consistent with the blowing snow model results of Dery and Yau (2002)
and the field observations of Mann et al., (2000). A plot of the average particle density for the
blowing snow layer in Fig. 1 is shown in Fig. 2.



Once an estimate of blowing snow particle number density and radii are obtained, the
sublimation rate of the particles can be computed based on the theoretical knowledge of the
process. Following Dery and Yau, (2002), the blowing snow mixing ratio $q_b$ (Kg ice / Kg air) is
given by:
$\qquad q_b(z) = \dfrac{4\pi\,\rho_{ice}\,r^3(z)\,N(z)}{3\,\rho_{air}} \qquad\qquad\qquad (3)$
Or substituting for $N(z)$ (Eq. 2):
$\qquad q_b(z) = \dfrac{2\,\rho_{ice}\,r(z)\,[\beta(z) - \beta_m(z)]\,S}{3\,\rho_{air}} \qquad\qquad\qquad (4)$
Where $\rho_{ice}$ is the density of ice (917 kg m$^{-3}$), and $\rho_{air}$ the density of air. Again following Dery and
Yau (2002) and others, the sublimation $S_b$ at height $z$ is computed from:
$\qquad S_b(z) = \dfrac{q_b(z)\,N_u\,[q_v(z)/q_{is}(z) - 1]}{2\,\rho_{ice}\,r^2(z)\,[F_k(z) + F_d(z)]} \qquad\qquad\qquad (5)$
Or, letting $\alpha(z)$ be the extinction and substituting for $q_b(z)$:
$\qquad S_b(z) = \dfrac{\alpha(z)\,N_u\,[q_v(z)/q_{is}(z) - 1]}{3\,\rho_{ice}\,r(z)\,[F_k(z) + F_d(z)]} \qquad\qquad\qquad (6)$
Where Nu is the Nusslet number defined as: $Nu = 1.79 + 0.606\,Re^{0.5}$
with the Reynolds number being: $Re = 2r(z)\,v_b/v$
where $v_b$ is the snow particle fall speed (assumed here to be 0.1 ms$^{-1}$) and $v$  the kinematic
viscosity of air (1.512x10$^{-5}$ m$^2$s$^{-1}$). $q_v$ is the water vapor mixing ratio of the air (obtained from
model data), $q_{is}$ is the saturation mixing ratio with respect to ice, and $F_k$ and $F_d$ are the heat
conduction and diffusion terms (m s Kg$^{-1}$):
$\qquad F_k = \left(\dfrac{L_s}{R_v T} - 1\right)\dfrac{L_s}{KT}$

$\qquad\quad F_d = \dfrac{R_v T}{D\,e_i(T)}$





Where $L_s$ is the latent heat of sublimation ($2.839 \times 10^6$ J/Kg), $R_v$ is the individual gas constant for
water vapor (461.5 J $Kg^{-1}$ $K^{-1}$), T is temperature (K), $K$ is the thermal conductivity of air, and $D$
the coefficient of diffusion of water vapor in air (both $D$ and $K$ are functions of temperature (see
Rogers and Yau, 1989). $S_b$ has units of Kg $Kg^{-1}$ $s^{-1}$. This can be interpreted as the mass of snow
sublimated per mass of air per second.
Then the column integrated blowing snow sublimation is:
$\qquad Q_s = \rho_{air} \ \int_{z=0}^{Z_{top}} S_b(z) \ dz$ (7)
Where $Z_{top}$ is the top of the blowing snow layer and $dz$ is 30 meters. $Q_s$ has units of Kg $m^{-2}$ $s^{-1}$).
Conversion to mm snow water equivalent (swe) per day is performed by multiplying by a
conversion factor:
$\qquad \rho' = 10^3 \ N_s / \rho_{ice}$ (8)
Where $N_s$ is the number of seconds in a day (86,400). The total sublimation amount in mm swe
per day is then:
$\qquad Q' = \rho' \ Q_s$ (9)
This computation is performed for every blowing snow detection along the CALIPSO track over
Antarctica. A 1 x 1 degree grid is then established over the Antarctic continent and each
sublimation calculation ($Q'$) is added to its corresponding grid box over the length of time being
considered (i.e. a year or month). This value is then normalized by the total number of CALIPSO
observations that occurred for that grid box over the time span. The total number of
observations includes all CALIPSO shots within the grid box for which a ground return was
detected, regardless of whether blowing snow was detected for that shot or not. Thus, the
normalization factor is the total number of shots with ground return detected for that box and
is always greater than the number of blowing snow detections (which equals the number of
sublimation retrievals). In order for the blowing snow detection algorithm to function, it must
first detect the position of the ground return in the backscatter profile. If it cannot do so, it is



not considered an observation. Over the interior of Antarctica, failure to detect the surface
does not occur often as cloudiness is less than 10 % and most clouds are optically thin. Near the
coasts, optically thick clouds become more prevalent. This approach will result in higher
sublimation values for those grid boxes that contain a lot of blowing snow detections and vice
versa (as opposed to just taking the average of the sublimation values for a grid box).

**2.2 Transport**
The transport of blowing snow is computed using the CALIPSO retrievals of blowing snow
mixing ratio and the MERRA-2 winds. A transport value is computed at each 30 m bin level and
integrated through the depth of the blowing snow layer:
$$Q_t = \rho_{air} \int_{z=0}^{Z_{top}} q_b(z)\ u(z)\ dz \tag{10}$$
Where $q_b(z)$ is the blowing snow mixing ratio from Eq. (3) and $u(z)$ is the MERRA-2 wind speed
at height $z$ and $Q_t$ has units of kg m$^{-1}$ s$^{-1}$. The wind speed is linearly interpolated from the
nearest two model levels. As with the sublimation, these values are gridded and normalized by
the total number of observations. The transport values are computed for each month of the
year by summing daily values and then multiplying by the number of seconds in the month
(resulting units of kg m$^{-1}$). The monthly values are then summed to obtain a yearly amount. A
further conversion is performed to produce units of Gt m$^{-1}$ yr$^{-1}$ by dividing by 10$^{12}$ (1000 kg per
metric ton and 10$^9$ tons per Gt).
**3 Results**
**3.1 Sublimation**
Figure 3 shows the average blowing snow frequency and corresponding total annual blowing
snow sublimation over Antarctica for the period 2007–2015. The highest values of sublimation
are along and slightly inland of the coast. Notice that this is not necessarily where the highest



blowing snow frequencies are located. Sublimation is highly dependent on the air temperature
and relative humidity. For a given value of the blowing snow mixing ratio ($q_b$), the warmer and
drier the air, the greater the sublimation. In Antarctica, it is considerably warmer along the
coast but one would not necessarily conclude that it is drier there. However, other authors
have noted that the katabatic winds, flowing essentially downslope, will warm and dry the air
as they descend (Gallee, 1998, and others). We have examined the MERRA-2 relative humidity
(with respect to ice) and indeed, according to the model, it is usually drier along the coast. The
model data often shows 90 to 100 % (or even higher) relative humidity for interior portions of
Antarctica, while along the coast it is often 60 % or less. It should be noted, however, that this
model prediction has never been validated through observations. The combination of warmer
and drier air makes a big difference in the sublimation as shown in Fig. 4. For a given relative
humidity the sublimation can increase by almost a factor of 100 as temperature increases from
-50 to -10 °C. For temperatures greater than -20 °C, sublimation is very dependent on relative
humidity, but this dependence lessens somewhat at colder temperatures. Continental interior
areas with very high blowing snow frequency that approach 75 % (like the Mega Dune region in
East Antarctica) exhibit fairly low values of sublimation because it is very cold and the model
relative humidity is high.
Figure 5 shows the annual total sublimation for years 2007–2015. It is evident that the
sublimation pattern or magnitude does not change much from year to year. The overall spatial
pattern of sublimation is similar to the model prediction of Dery and Yau, (2002) with our
results showing noticeably greater amounts in the Antarctic interior and generally larger values
near the coast. As previously noted, most sublimation occurs near the coast due mainly to the
warmer temperatures. The areas of sublimation maximums near the coast are consistently in
the same location year to year, indicating that these areas may experience more blowing snow
episodes and possibly more precipitation (availability of snow to become airborne). It is
interesting to compare the sublimation pattern with current estimates of Antarctic
precipitation. Precipitation is notoriously difficult to quantify over Antarctica due to the scarcity
of observations and strong winds producing drifting and blowing snow which can be
misidentified as precipitation. Precipitation is often measured by looking at ice cores or is



estimated by models. But perhaps the most complete (non-model) measure of Antarctic
precipitation come from the CloudSat mission. Palerme et al., (2014) used CloudSat data to
construct a map of Antarctic precipitation over the entire continent (north of 82 S). They
showed that along the East Antarctic coast and slightly inland, precipitation ranges from 500 to
700 mm swe yr$^{-1}$ and decreases rapidly inland to less than 50 mm yr$^{-1}$ in most areas south of 75
S. Their precipitation pattern is in general agreement with the spatial pattern of our
sublimation results and the magnitude of our sublimation estimates is in general less than the
precipitation amount, with a few exceptions. These occur mostly inland in regions of high
blowing snow frequency such as the Megadune region and in the general area of the Lambert
glacier. In these regions, our sublimation estimates exceed the CloudSat yearly precipitation
estimates. When this occurs, it is likely that either the precipitation estimate is low or the
sublimation estimate is too high. Otherwise it would indicate a net negative mass balance for
the area unless transport of snow into the region accounted for the difference.
Table 1 shows the average sublimation over all grid cells in snow water equivalent and the
integrated sublimation amount over the Antarctic continent (north of 82S) for the CALIPSO
period in Gt yr$^{-1}$. Note that the 2006 data include only months June–December (CALIOP began
operating in June, 2006) and the 2016 data are only up through October, and do not include the
month of February (CALIOP was not operating). To obtain the integrated amount, we take the
year average swe (column 1) multiplied by the surface area of Antarctica north of 82S and the
density of ice. The average integrated value for the 9 year period 2007–2015 of 419 Gt yr$^{-1}$ is
significantly greater than (about twice) values in the literature obtained from model
parameterizations (Lenaerts 2012b). Note also that these figures do not include the area
poleward of 82S, the southern limit of CALIPSO observations. If included, and the average
sublimation rate over this area were just 4 mm swe per year, this would increase the
sublimation total by 10 Gt yr$^{-1}$. Gallee (1998) used a model to compute blowing snow
sublimation over Antarctica and obtained a continent-wide average value of 0.087 mm per day.
Further he postulated that If this were constant over a year, it would amount to about 32 mm
swe and if all of it were deposited in the ocean it would account for 1.3 mm yr$^{-1}$ of sea level rise.
The same logic applied to our results would yield a 1.7 mm yr$^{-1}$ seal level rise.



Our results (Table 1) show somewhat of a decreasing trend to the data with the years 2007 and
2008 having the highest sublimation values while 2014 and 2015 have the lowest. If this trend is
real, it is not clear what is causing it. The frequency of blowing snow is relatively constant over
this period when integrated over the continent (not shown). Other factors that could cause a
decrease in sublimation are colder temperatures, increased humidity, or decreased blowing
snow particle concentration (which would equate to a decreased average layer integrated
backscatter).
Palerme et al., (2014) has shown that the mean snowfall rate over Antarctica (north of 82 S)
from August 2006 to April 2011 is 171 mm yr$^{-1}$. The average yearly snow water equivalent
sublimation from Table I is the average sublimation over the continent (also north of 82 S). For
the same time period, our computed CALIPSO-based average blowing snow sublimation is
about 50 mm yr$^{-1}$. This means that on average, over one third of the snow that falls over
Antarctica is lost to sublimation through the blowing snow process. In comparison surface
sublimation (sublimation of snow on the surface) is considered to be relatively small (about a
tenth of airborne sublimation) except in summer (Lenearts 2012a, 2012b).
**3.2 Transport**
Transport of snow via the wind is generally important locally and does not constitute a large
part of the ice sheet mass balance in Antarctica. There are areas where the wind scours away all
snow that falls producing a net negative mass balance (i.e. blue ice areas), but in general, the
snow is simply moved from place to place over most of the continent. At the coastline,
however, this is not the case. There, persistent southerly winds can carry airborne snow off the
continent. This can be seen very plainly in Fig. 6 which is a MODIS false color (RGB = 2.1, 2.1, .85
μm) image of a large area of blowing snow covering an area about the size of Texas in East
Antarctica. We have found this false color technique to be the best way to visualize blowing
snow from passive sensors. The one drawback is that sunlight is required. In the figure, blowing
snow shows up as a dirty white, the ice/snow surface (in clear areas) is blue and clouds are
generally a brighter white. Also shown in Fig. 6 are two CALIPSO tracks (yellow lines) and their



associated retrieved blowing snow backscatter (upper and lower images of CALIOP
backscatter). Note that the yellow track lines are drawn only where blowing snow was detected
by CALIOP and that not all the CALIOP blowing snow detections are shown. The green dots
denote the coastline. Plainly seen along the coast near longitude 145–150E is blowing snow
being carried off the continent. In this case, topography might have played a role to funnel the
wind in those specific areas. Figure 7 shows a zoomed in image of this area with the red lines
indicating the approximate position of the coastline. Also note that, as evidenced by the times
of the MODIS images, this transport began on or before October 13 at 23:00 UTC and continued
for at least 7 hours. This region is very close to the area of maximum sublimation seen in Fig. 3
and shown to be quite stable from year to year in Fig. 5. Undoubtedly, this continent to ocean
transport also occurs in other coastal areas of Antarctica and most often during the dark winter
(when MODIS could not see it).
In an attempt to better understand the magnitude of this phenomena, we have computed the
amount of snow mass being blown off the continent by computing the transport at 342 points
evenly spaced (about 70 km apart) along the Antarctic coast using only the v component of the
wind. If the v component is positive, then the wind is from south to north. The transport (Eq.
(10) using only the v wind component) is computed at each coastal location and then summed
over time at that location. The resulting transport is then summed over each coastal location to
arrive at a continent-wide value of transport from continent to ocean. Of course this assumes
that the coastline is oriented east-west everywhere. This is true of a large portion of Antarctica
but there are regional exceptions. Thus we view the results shown in Table II to be an upper
limit of the actual continent to ocean transport. Evident from Table 2 is that most of the
transport for East Antarctica occurs in a relatively narrow corridor, with on average over half
(51 %) of the transport occurring between 135E and 160E. This is obviously due to the very
strong and persistent southerly winds (see Fig. 10) and high blowing snow frequency in this
region and is consistent with the conclusions of Scarchilli et al., (2010). In West Antarctica, an
even greater fraction (60 %) of the transport off the coast occurs between 80W and 120W.



In Fig. 8 we show the magnitude of blowing snow transport for the 2007–2015 timeframe in Mt
km$^{-1}$ yr$^{-1}$ as computed from Eq. (10). The magnitude of snow transport, as expected, closely
resembles the overall blowing snow frequency pattern as shown in Fig. 3. The maximum values
(white areas in Fig. 8) exceed about 3 x 10$^6$ tons of snow per km per year. In Figs. 9 and 10 we
display the MERRA-2 average 10 m wind speed and direction for the years 2007–2015. By
inspection of Figs. 8 and 10 it is seen that the overall transport in East Antarctica is generally
from south to north and obviously dominated by the katabatic wind regime. It is immediately
apparent that the average wind speed and direction does not change much from year to year,
with the former helping to explain why the average continent-wide blowing snow frequency is
also nearly constant from year to year (not shown).
**4. Error Analysis**
There are a number of factors that can affect the accuracy of the results presented in this work.
These include:
1) Error in the calibrated backscatter and conversion to extinction
2) Errors in the assumed size of blowing snow particles
3) Not correcting for possible attenuation above the blowing snow layer
4) Misidentification of some layers as blowing snow when in fact they were not (false positives)
5) Failure to detect some layers (false negatives)
6) Errors in the MERRA-2 temperature and moisture data
7) Limited spatial sampling
The magnitude of some of these can be estimated, others are hard to quantify. For instance, 1),
2) and 6) are directly involved in the calculation of sublimation (Eq. 6). The error in extinction,
particle radius, temperature and moisture can be estimated. The error associated with 3) is
probably very small over the interior of Antarctica, but could be appreciable nearer the
coastline. In the interior, clouds are a rare occurrence and when present are usually optically
thin. Cloudiness increases dramatically near the coast both in terms of frequency and optical
depth. Here the effect of overlying attenuating layers could be appreciable in that it would



reduce the backscatter of the blowing snow layer and the derived extinction. This in turn would
lead to a lower blowing snow mixing ratio and thus lower sublimation and transport.
There is a further point to be made with respect to clouds that relates to 5) above. The method
we use to detect blowing snow will not work in the presence of overlying, fully attenuating
clouds. It is reasonable to suspect that cyclonic storms which impinge upon the Antarctic coast
and travel some distance inland would be associated with optically thick clouds and contain
both precipitating and blowing snow. Our method would not be able to detect blowing snow
during these storms, but we would not count such cases as "observations", since the ground
would not be detected. The point is, blowing snow probably occurs often in wintertime
cyclones, but we are not able to detect it. This could lead to an under prediction of blowing
snow occurrence, especially near the coast. Also, blowing snow layers less than 20 - 30 m thick
would also likely be missed. It is not clear how often these layers occur, but they are known to
exist and missing them will produce an underestimate of blowing snow sublimation and
transport amounts. With regard to spatial sampling (7 above), unlike most passive sensors,
CALIPSO obtains only point measurements along the spacecraft track at or near nadir. On a
given day, sampling is poor. CALIPSO can potentially miss a large portion of blowing snow
storms such as is evidenced from inspection of Fig. 6. We have seen many examples of such
storms in both the MODIS and CALIPSO record. Quantifying the effect of poor sampling on
sublimation estimates would be difficult but should be pursued in future work.
In an effort to quantify the error in our sublimation estimates, here we assume the extinction
error to be 20 %, the particle radius error 10 % and the temperature and moisture error 5 %
each. In Eq. (6) these terms are multiplicative. The total error in sublimation is then:
$\pm 1 - (0.8 * 0.9 * 0.95 * 0.95) = \pm 0.35$
This indicates that the sublimation values derived in this work should be considered to have an
error bar of ±35 %. The error in computed transport involves error in wind speed and the
blowing snow mixing ratio, the latter being dependent on extinction and particle size. If we



assume wind speed has an error of 20 %, extinction 20 % and particle size 10 %, the total error
in transport is:
±1 – (0.8 * 0.8 * 0.9) = ±0.42
**5. Summary and Discussion**
This paper presents the first estimates of blowing snow sublimation and transport over
Antarctica that are based on actual observations of blowing snow layers from the CALIOP space
borne lidar. We have used the CALIOP blowing snow retrievals combined with MERRA-2 model
re-analyses of temperature and moisture to compute the temporal and spatial distribution of
blowing snow sublimation and transport over Antarctica for the first time. The results show that
the maximum sublimation, with annual values exceeding 250 mm swe, occurs within roughly
200 km of the coast even though the maximum frequency of blowing snow most often occurs
considerably further inland. This is a result of the warmer and drier air near the coast (at least
in the MERRA-2 re-analysis data) which substantially increases the sublimation. In the interior,
extremely cold temperatures and high model relative humidity lead to greatly reduced
sublimation. However, the values obtained in parts of the interior (notably the Megadune
region of East Antarctica - roughly 75 to 82S and 120 to 160E) are considerably higher than
prior model estimates of Dery and Yau (2002) or Lenaerts et al., 2012a). This is most likely due
to the very high frequency of occurrence of blowing snow as detected from CALIOP data in this
region which is not necessarily captured in models (Lenaerts et al., 2012b).
The overall spatial pattern of blowing snow sublimation is consistent with previous modelling
studies (Dery and Yau, 2002 and Lenearts et al., 2012a). However, we find the Antarctic
continent-wide integrated blowing snow sublimation to be about twice the amount of previous
studies such as Lenaerts et al., (2012a) (393 ±138 vs roughly 190 Gt yr$^{-1}$), even though the
observations include only the area north of 82° S. The maximum in sublimation is about 250
mm swe per year near the coast between longitudes 140E and 150E and seems to occur
regularly throughout the 11 year data record. One reason for the higher sublimation values in
this study is that the average blowing snow layer depth as determined from the CALIOP




measurements is 120 m. Layers as high as 200 - 300 m are not uncommon. It is likely that
models such as those cited above do not always capture the full depth of blowing snow layers,
thus producing a smaller column-integrated sublimation amount. Another reason for
differences between our result and previous model estimates lies in the fact that we only
compute sublimation from blowing snow layers that are known to exist (meaning they have
been detected from actual backscatter measurements). Models, on the other hand, must infer
the presence of blowing snow from pertinent variables within the model. The existence of
blowing snow is not easy to predict. It is a complicated function of the properties of the
snowpack, surface temperature, relative humidity and wind speed. Snowpack properties
include the dendricity, sphericity, grain size and cohesion, all of which can change with the age
of the snow. In short, it is very difficult for models to predict exactly when and where blowing
snow will occur, much less the depth that blowing snow layer will attain.
The spatial pattern of the transport of blowing snow follows closely the pattern of blowing
snow frequency. The maximum transport values are about 5 Megatons per km per year and
occur in the Megadune region of East Antarctica with other locally high values at various
regions near the coast that generally correspond to the maximums in sublimation. We
attempted to quantify the amount of snow being blown off the Antarctic continent by
computing the transport along the coast using only the v component of the wind. While this
may produce an overestimate of the transport (since the Antarctic coast is not oriented east-
west everywhere), we find the amount of snow blown off the continent to be significant and
fairly constant from year to year. The average off-continent transport for the 9 year period
2007–2015 was 3.68 Gt yr$^{-1}$ with about two thirds of that coming from East Antarctica and over
one third from a relatively small area between longitudes 135E and 160E.
Over the nearly 11 years of data, the inter-annual variability of continent wide sublimation
(Table 1) can be fairly large – 10 to 15 % - and likely the result of precipitation variability. There
seems to be a weak trend to the sublimation data with earlier years having greater sublimation
than more recent years. However, based on the likely magnitude of error in the sublimation
estimates, the trend cannot be considered statistically significant.



**Data Availability**

The CALIPSO calibrated attenuated backscatter data used in this study can be obtained from the NASA Langley Atmospheric Data Center at: https://earthdata.nasa.gov/about/daacs/daac-asdc. CALIPSO Science Team (2015), CALIPSO/CALIOP Level 1B, Lidar Profile Data, version 3.30, Hampton, VA, USA: NASA Atmospheric Science Data Center (ASDC), Accessed at various times during 2016 at doi: 10.5067/CALIOP/CALIPSO/CAL_LID_L1-ValStage1-V3-30_L1B-003.30

The MERRA-2 data are available from the Goddard Earth Sciences Data and Information Services Center (GESDISC) at: https://disc.gsfc.nasa.gov/datareleases/merra_2_data_release.

The blowing snow data (layer backscatter, height, etc.) are available through the corresponding author and will be made publicly available through the NASA Langley Atmospheric Data Center in the near future.

**Acknowledgements**

This research was performed under NASA contracts NNH14CK40C and NNH14CK39C. The authors would like to thank Dr Thomas Wagner and Dr David Considine for their support and encouragement. The CALIPSO data used in this study were the DOI: 10.5067/CALIOP/CALIPSO/LID_L1-ValStage1-V3-40_L1B-003.40 data product obtained from the NASA Langley Research Center Atmospheric Science Data Center. We also acknowledge the Global Modeling and Assimilation Office (GMAO) at Goddard Space Flight Center who supplied the MERRA-2 data and the AMPS data were kindly supplied by Julien Nicolas of the Byrd Polar and Climate Research Center at Ohio State University.

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





Table I. The year average sublimation per year (average off all grid boxes) and the integrated
sublimation over the Antarctic continent (north of 82S). [*]2006 and 2016 consist of only 7 and 9
months of observations, respectively.

| Year | Average Sublimation (mm swe) | Integrated Sublimation (Gt yr$^{-1}$) |
|---|---|---|
| 2006[*] | 28.3 | 255 |
| 2007 | 56.8 | 514 |
| 2008 | 49.2 | 446 |
| 2009 | 45.3 | 409 |
| 2010 | 42.9 | 388 |
| 2011 | 47.6 | 431 |
| 2012 | 44.4 | 402 |
| 2013 | 47.7 | 432 |
| 2014 | 41.5 | 376 |
| 2015 | 41.3 | 374 |
| 2016[*] | 33.2 | 301 |
| **AVG** | **43.5[*]** | **393.4[*]** |


Table II. The total transport (Gt yr$^{-1}$) from continent to ocean for various regions in Antarctica
for 2007–2015.

| Year | East Antarctica | West Antarctica | 135E – 160E | 80W – 120W |
|---|---|---|---|---|
| 2007 | 2.52 | 1.29 | 1.72 | 0.82 |
| 2008 | 2.20 | 1.43 | 1.21 | 0.90 |
| 2009 | 2.63 | 1.27 | 1.51 | 0.78 |
| 2010 | 2.26 | 1.15 | 1.38 | 0.73 |
| 2011 | 2.04 | 1.04 | 1.13 | 0.64 |
| 2012 | 2.49 | 1.21 | 1.41 | 0.73 |
| 2013 | 2.54 | 1.41 | 1.26 | 0.83 |
| 2014 | 2.55 | 1.02 | 1.49 | 0.67 |
| 2015 | 2.76 | 1.38 | 1.58 | 0.69 |
| **Avg** | **2.44** | **1.24** | **1.41** | **0.75** |


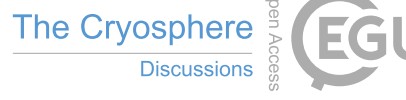



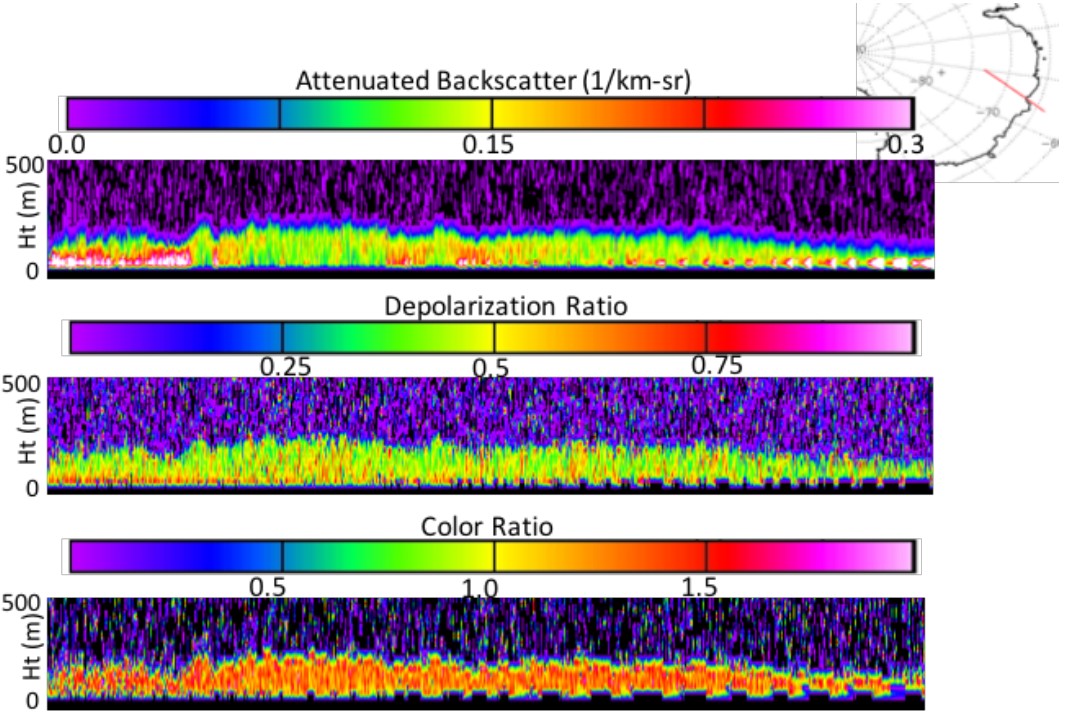


Figure 1. A typical Antarctic blowing snow layer as measured by CALIPSO on May 28, 2015 at
17:08:41 – 17:11:33 UTC. Displayed (from top to bottom) are the 532 nm calibrated, attenuated
backscatter, the depolarization ratio at 532 nm, and the color ratio (1064 nm / 532 nm).



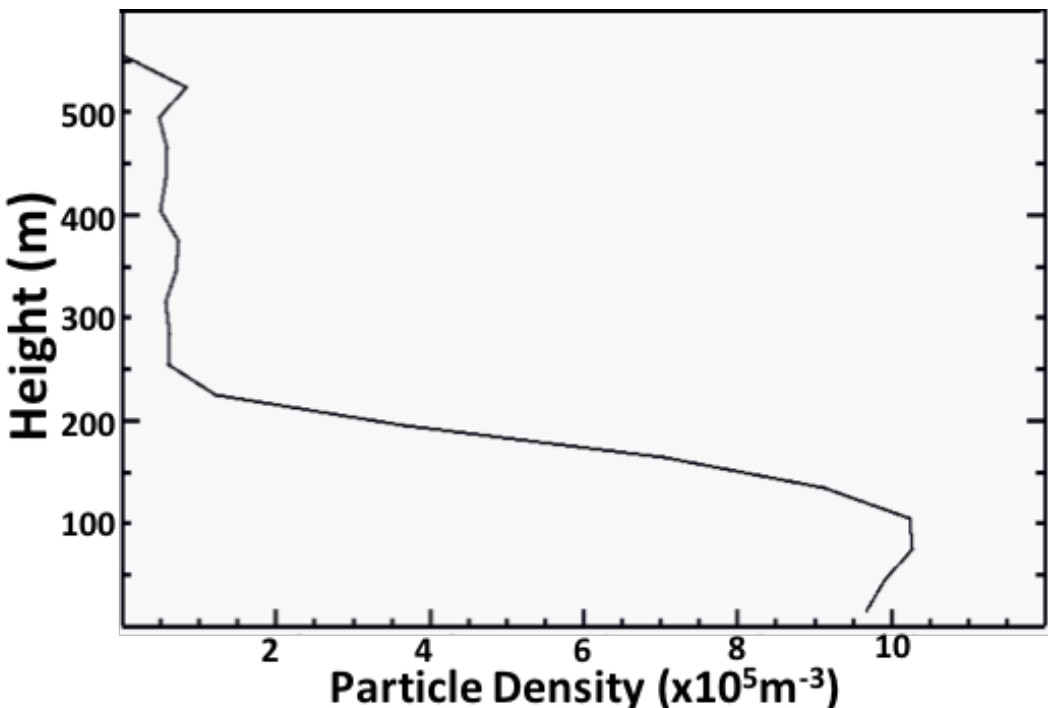


Figure 2. Average particle density profile (Eq. 2)through the blowing snow layer shown in Fig. 1.



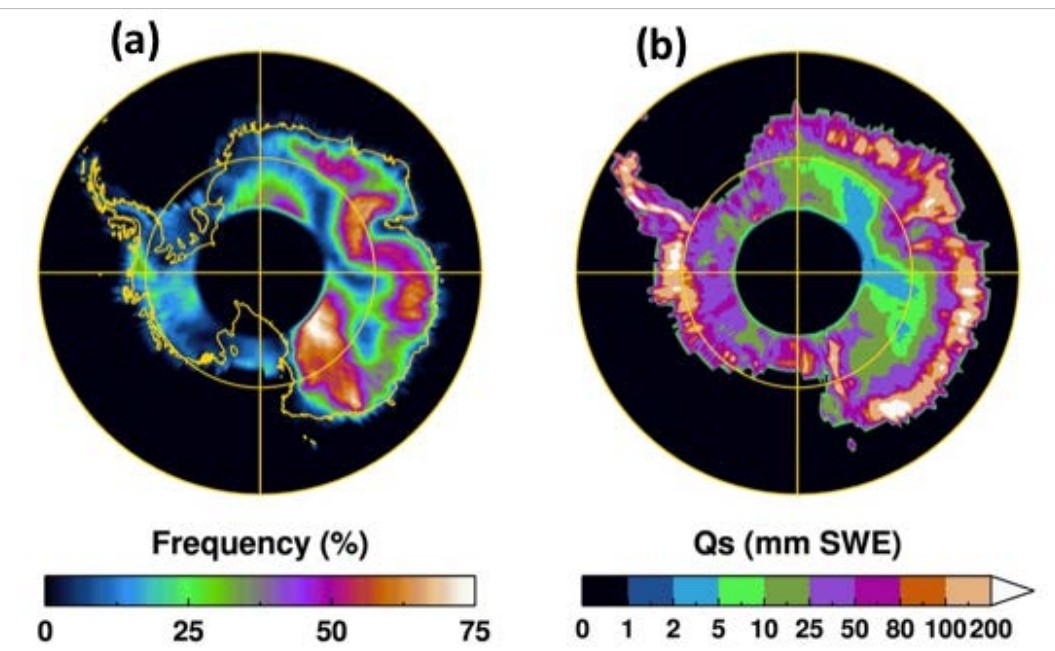


Figure 3. (a) The average April through October blowing snow frequency for the period 2007–

2015. (b) The average annual blowing snow sublimation for the same period as in (a).



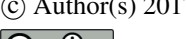

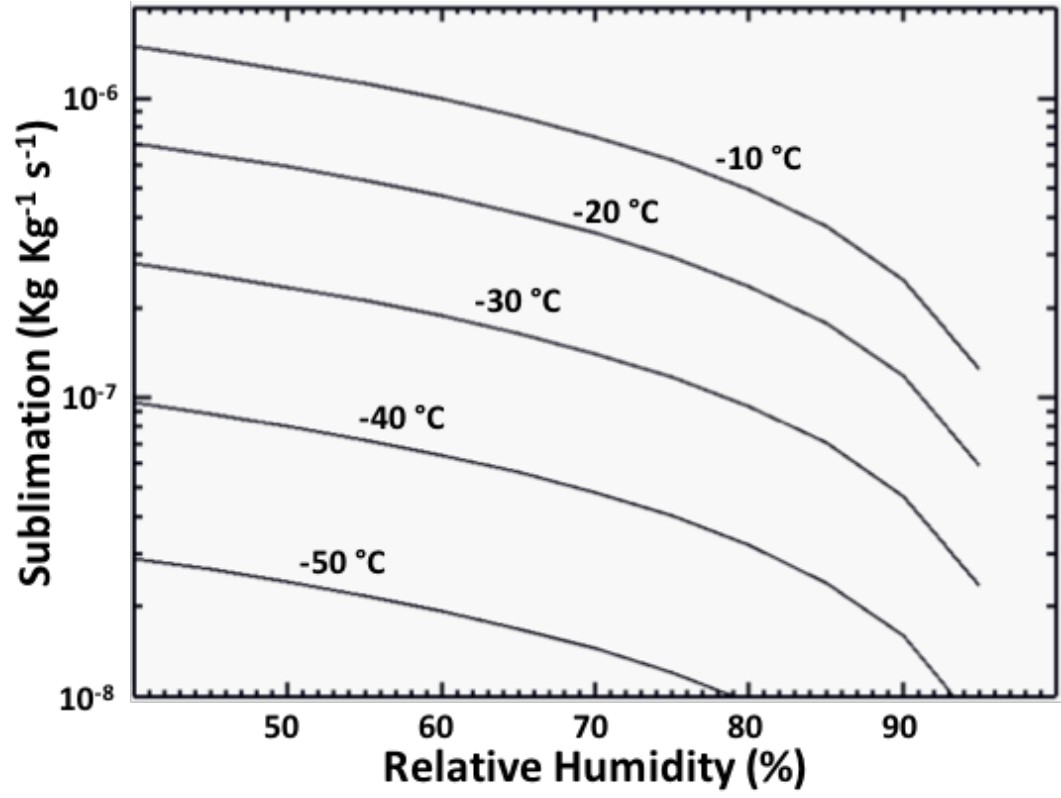


Figure 4. Computed blowing snow sublimation rate using Eqs. (3) and (4) as function of relative

humidity for varying air temperatures. The particle density value used in Eq. (3) was 106 m-3

which corresponds to a blowing snow mixing ratio (qb) of 4.7x10-5 Kg Kg-1





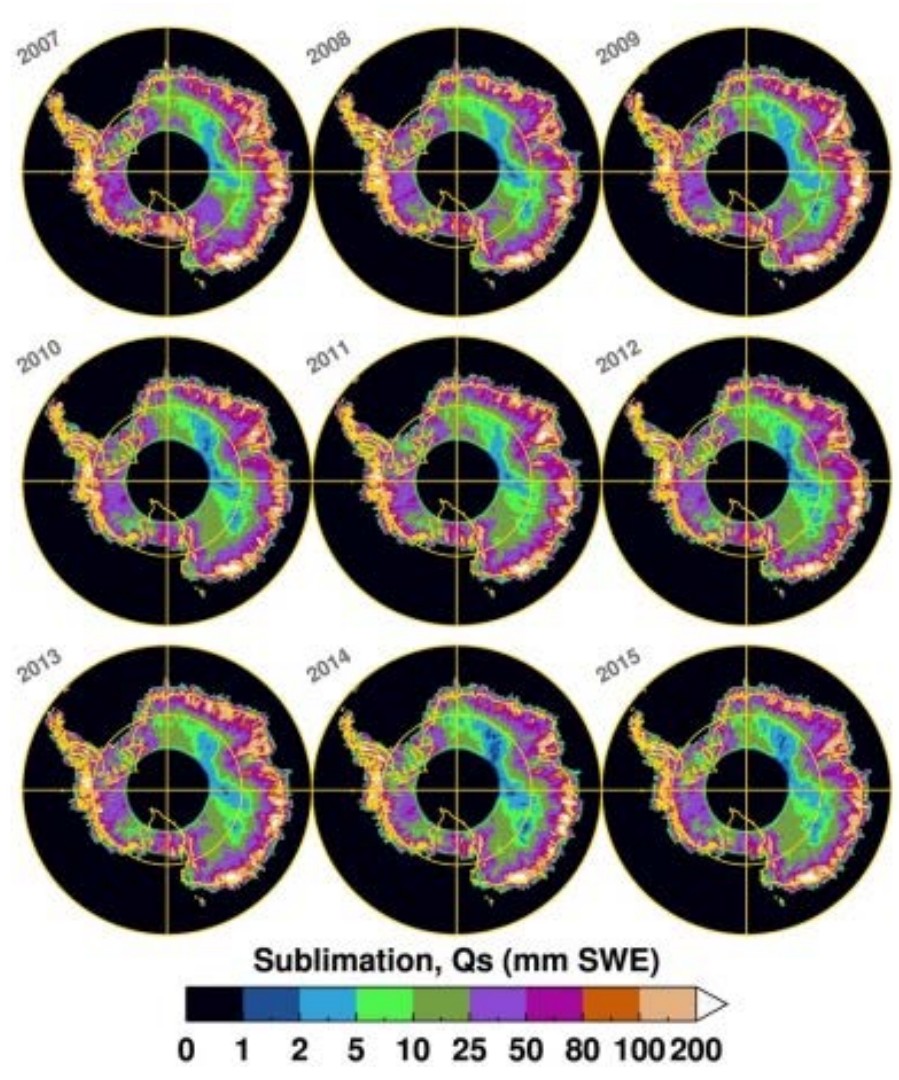


Figure 5. Blowing snow total sublimation over Antarctica by year for 2007–2015.





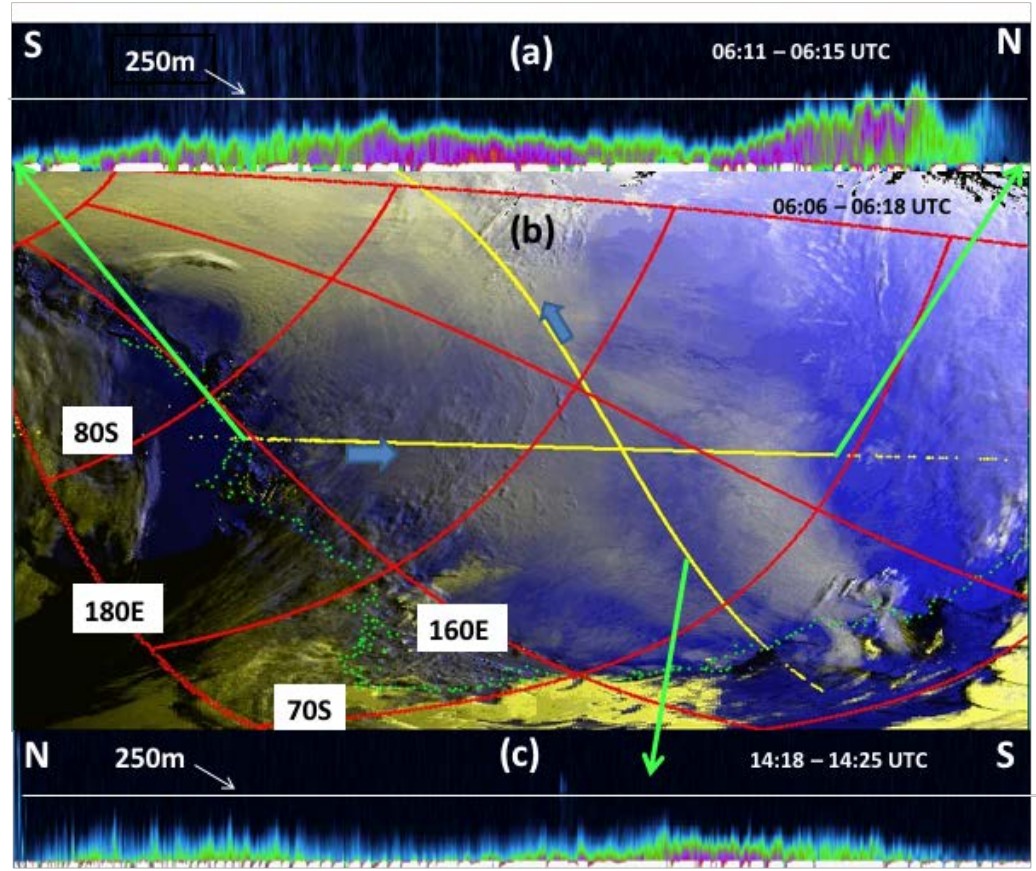


Figure 6. A large blowing snow storm over Antarctica with blowing snow transport from continent to ocean on October 14, 2009. (a) CALIOP 532 nm attenuated backscatter along the yellow (south to north) line bounded by the green arrows as shown in (b) at 06:11 – 06:15 UTC. (b) MODIS false color image at 06:06:14 – 06:17:31 UTC showing blowing snow as dirty white areas. The coastline is indicated by the green dots, and two CALIPSO tracks, where blowing snow was detected are indicated by the yellow lines. (c) CALIOP 532 nm attenuated backscatter along the yellow (north to south) line, 14:18 – 14:25 UTC.



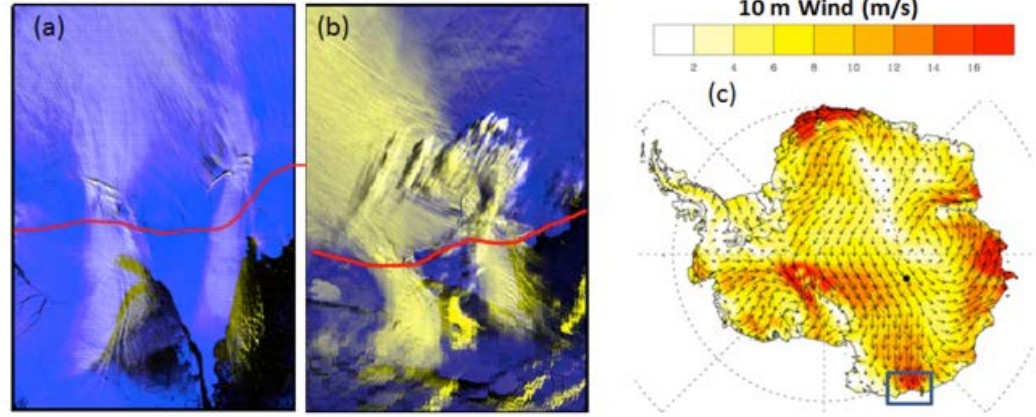


Figure 7. (a) MODIS false color image on October 13, 2009, 23:00 UTC  and (b) October 14,
2009, 06:16 UTC. The red line is the approximate position of the coastline. (c) The 10 m wind
speed from the AMPS model (Antarctic Mesoscale Prediction System) for October 14, 2009. The
area covered by the MODIS images is roughly that indicated by the blue box in (c).





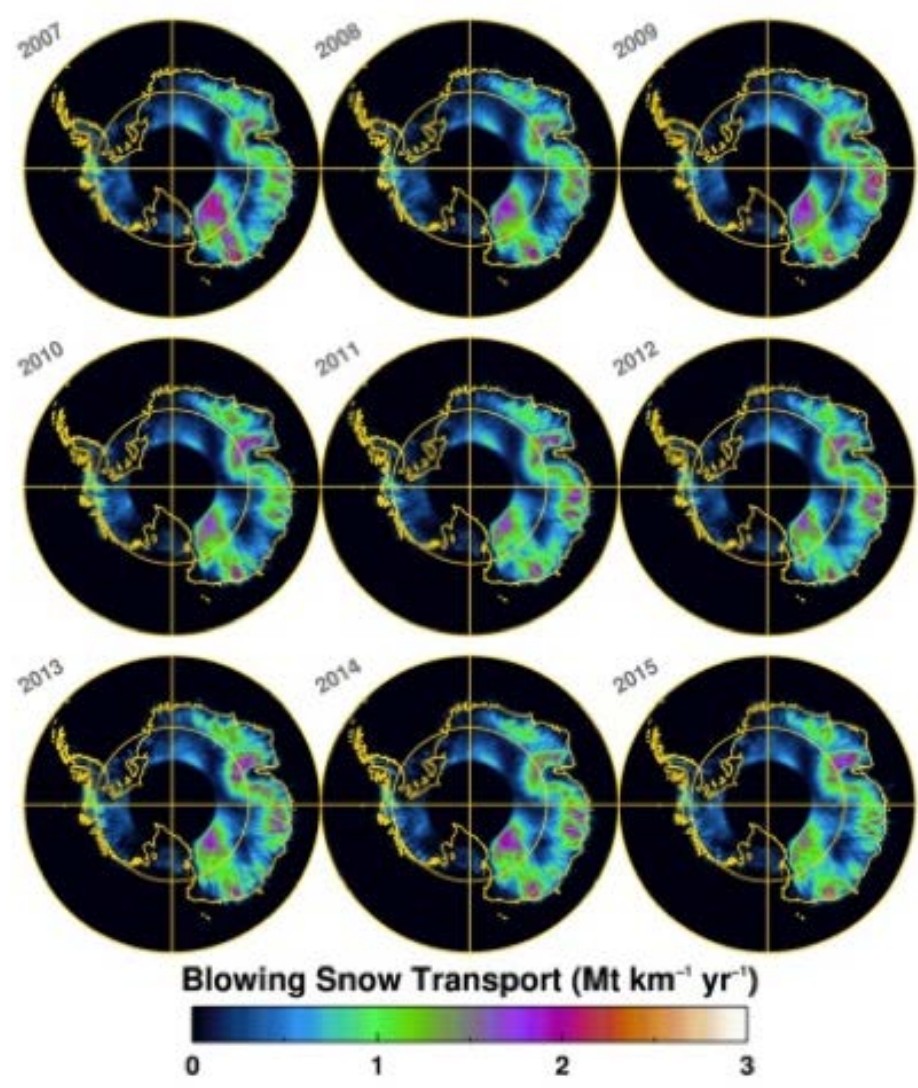


Figure 8. The magnitude of blowing snow transport over Antarctica integrated over the year for years 2007–2015.





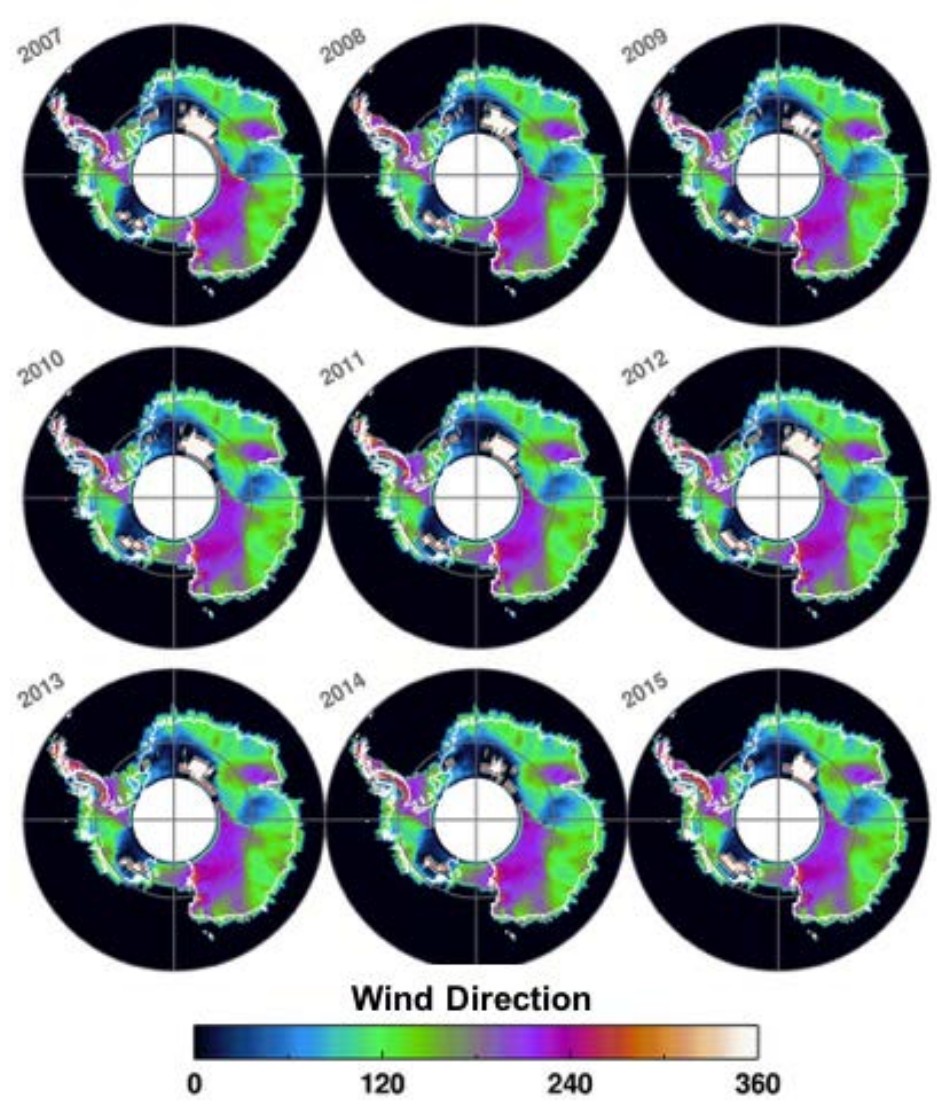


679        Figure 9. The average wind direction over Antarctica for years 2007–2015





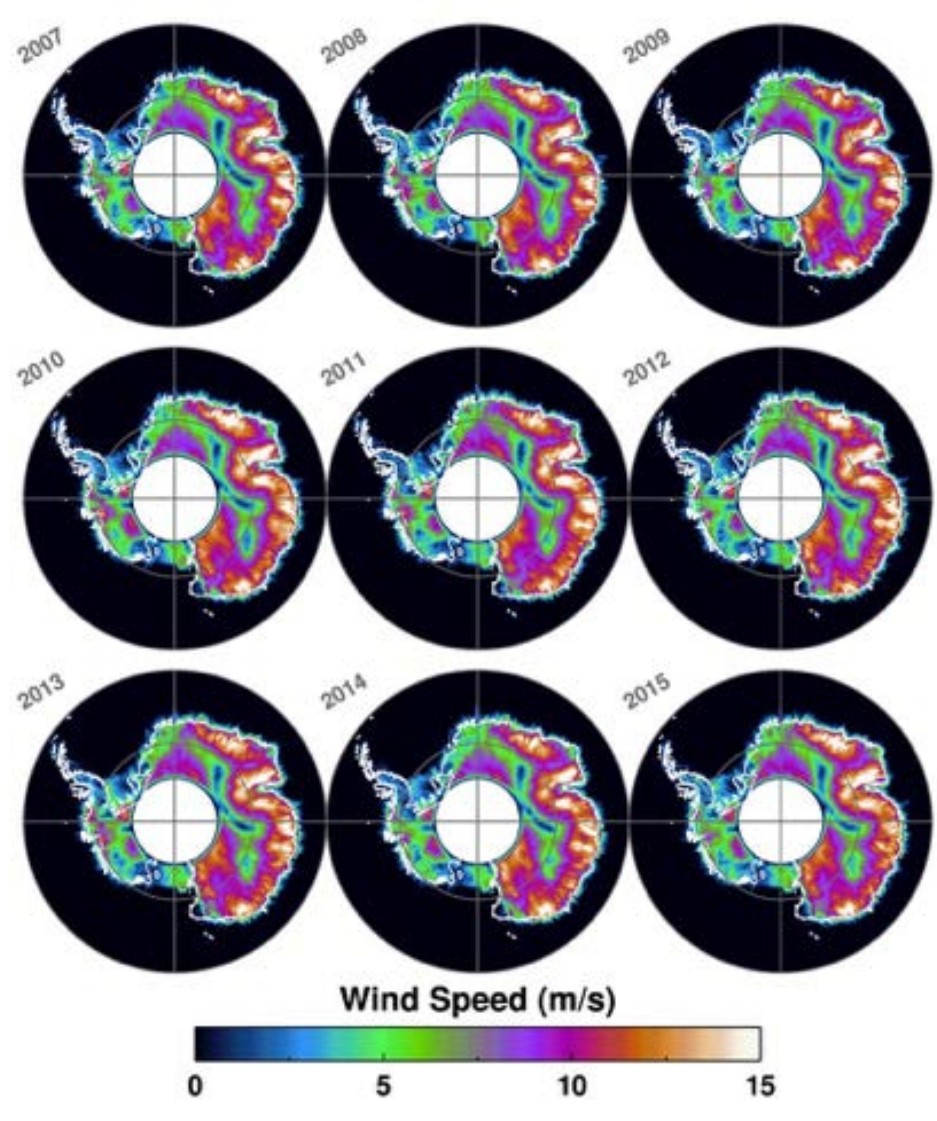


Figure 10. The average wind speed over Antarctica for years 2007–2015