# Peer review of "Blowing Snow Sublimation and Transport over Antarctica from 11 Years of CALIPSO"

_The Cryosphere, 2017_

## Referee Comment (RC1) · J. Lenaerts (Referee) · 1 May 2017

Palm et al. use CALIPSO observations and simple parameterizations to estimate Antarctic ice sheet blowing snow sublimation and transport for the period 2005-2017. This is an interesting, innovative, and timely contribution to the (necessarily) growing body of literature on the impact of blowing snow on ice sheet SMB, contains the methods and results to sufficiently support the conclusions, and is relatively well written (although that needs to be improved in some places). I support publication in TC after one major issue is considered, and thorough rewriting is performed in a revised version.

Major issue

I have strong reservations with regard to the use of simplified, steady-state parameterizations and meteorological forcing, which put a strong constraint on the resulting blowing snow sublimation. The authors use MERRA-2 temperature and RH data to derive sublimation rates, but these MERRA-2 data are (a) not at all evaluated over Antarctica, and – more importantly – (b) since MERRA-2 does not simulate blowing snow, do not imply the atmospheric effects of the well-documented self-limiting behavior of blowing snow sublimation, in which sublimation will lead to latent heat release to the atmosphere at the top of the blowing snow layer, in turn cooling and moistening the atmosphere and limiting subsequent sublimation. This effect is expected to have a first-order negative effect on the sublimation (while retaining the blowing snow layer transport active, so this is not observable from space), and should – in some way or another – be included in this approach. I realize that the authors do not (and do not want to) utilize a model that includes this behavior, nor include blowing snow processes in MERRA-2. One option is to perform multiple sensitivity tests with gradually higher RHice values and lower temperatures, based on and in combination with a MERRA-2 near-surface climate evaluation at select stations over Antarctica.

These sensitivity tests should be combined with varying other important parameters to determine sensitivity. For instance, why did the authors choose a fall velocity of 0.1 m/s? I would strongly suggest to expand the Section 4 and include a detailed description of the sensitivity tests.

Minor issues

The writing should be improved in places and caution is warranted to very clearly describe the process the authors are referring to. Also, some parts are clearly too speculative and should be revised (see below).

L13: near-surface

L15: define surface mass balance

[Figure]

L17 and beyond: clearly mention the time period considered in this study

L23: blowing snow sublimation!

L29: 2006-2015

L94: it would be helpful to mention all sublimation rates (also those from earlier literature) in the same units to facilitate comparison. Which time period are these from?

L108: 20,000 km – reference needed

L152: 1064/532 – include units

L189: (Walden et al., 2003)

L233: How to go from blowing snow mixing ratio to extinction? What are the units of this extinction, and why does they relate as alfa(z) = 3/2 qb(z)/r(z) ?

L240 and around: The use of MERRA-2 needs to be described here. How are T and RH incorporated here? How is temporal and vertical interpolation dealt with?

L241: Equations are not numbered

L321: It would be very helpful to plot the CloudSAT precipitation numbers and plot the ratio sublimation/precipitation to guide this discussion.

L339: 419 Gt/yr – this is a different number than mentioned anywhere else.

L344-348: This is extremely speculative and contains the wrong translation from Gt/yr to mm sea level rise ($\sim$360 Gt = 1 mm SLE). Most of the sublimation is probably recycled on the ice sheet, and of course Antarctic SMB is positive and dominated by precipitation. Please remove.

L349-355: is the trend significant? Probably not, with significant inter-annual variability and only 10 years in the time series. If it is not significant, please remove. If it is significant, it would be useful to relate this to MERRA-2 T and RH averages.

L357: clarify if you consider the grounded or total (include ice shelves) ice sheet.

L371: size of Texas – quantify.

Table I: Is the average from 2006-2015 (the full years)? Clarify.

Figures 9 and 10 can be removed or moved to supplements. They do not contain any results that are necessary to be shown in a separate (main) figure.

---

## Short Comment (SC1) · 3 May 2017

General comments This paper presents very interesting and unprecedented continent-wide statistics of blowing snow over Antarctica from long-term satellite observations. These include estimations of blowing snow sublimation, a significant but poorly known component of the Antarctic surface mass balance. Such works are essential for evaluation of atmospheric models from which the total surface mass budget of the ice sheet can be estimated. However, there are some important missing aspects and information in the study that I would like to report here. Of particular concern is the method from which sublimation estimates are computed. One possibly very significant source of error is an underestimation of atmospheric moisture by MERRA-2: the method does

not take into account the fact that moisture from blowing snow sublimation is retained while air flows further through blowing snow regions, strongly reducing (or cutting if saturation is reached) any further sublimation of blowing snow downstream. MERRA-2 does not account for blowing snow sublimation, thus the method constantly resets air moisture to values for which blowing snow sublimation has never occurred, and very likely overestimates total sublimation.

Specific comments Observational studies on blowing snow in Antarctica are very scarce, to the extent that continuous measurements extending beyond a few weeks or months barely exist. However, considerable efforts have been made in the recent years on that specific topic, that you might have missed in your bibliography. An observation campaign dedicated to blowing snow has been run in January 2010 by the Laboratoire de Glaciologie et Géophysique de l'Environnement (LGGE, France) in coastal Adélie Land. Some of the collected data have been presented, for instance, in Trouvilliez et al. (2014), Barral et al. (2014) and Amory et al. (2016, 2017), and used for evaluation of preliminary modelling results (Gallée et al. 2013, Amory et al. 2015). Ground measurements on the ice sheet have been performed using second-generation acoustic FlowCapt™ sensors. While these sensors have been shown to slightly underestimate the blowing snow flux compared to optical snow particle counters SPC-S7 in the French Alps, they remain excellent detectors of blowing snow occurrences (Trouvilliez et al. 2015). To date, up to 7 years (2010-2016) of continuous ground measurements of blowing snow frequency in coastal Adélie Land are available (for comparison with CALIPSO data). The dataset also includes (discontinuous) measurements of snow particle size performed since 2013 at 50-m height above the ground with a SPC at Dumont d'Urville station (see Palerme 2014). I'm part of the research team that has produced (and still does) these observations and I'm open to discuss it with the authors if they wish.

P8, L222: Figure 2 shows an increase in particle density with height for the first 100 meters above the surface. This is surprising since the density of blowing snow particles

is supposed to decrease as distance from the ground (i.e., from the particle source) increases (see for instance the strong decrease within the first 10 meters above the ground in Fig. 4 of Mann et al. 2000). Have you an idea of what can cause this feature?

P17, L431: In addition, clouds may be associated with precipitation which contributes moistening the dry surface air layer (Grazioli et al. 2017; http://www.the-cryosphere-discuss.net/tc-2017-18/) and thus correspondingly reduces blowing snow sublimation.

P17, L449 and onwards: Although this aspect is already partly discussed in the paper, estimating blowing snow sublimation by using MERRA-2 re-analysis fields of moisture could be misleading because i) re-analysed moisture near the surface could be under-estimated and ii) no retro-action of sublimation on moisture is accounted for. Systematic dry biases in atmospheric models and meteorological (re-) analyses that do not account for blowing snow have been discussed in Barral et al. (2014). Using a 3-year dataset of ground measurements at a coastal location in Adélie Land, they showed (their Figure 6) for 3 modelling products that the moisture error in the near-surface layer for the continental grid point closest to the measurement location is much larger than 5% (as considered in the error analysis in Section 4), and that the 3 models fail to represent the observed increase of atmospheric moisture with wind speed. For instance, the moisture error almost averages 100% for the ECMWF operational analysis for wind speeds exceeding 12 m/s. It is likely that most meteorological and climate models ignoring blowing snow are affected by similar dry biases, at least over windy peripheral areas of East Antarctica where blowing snow is highly active. In addition, in the blowing snow layer the air quickly saturates as part of the blowing snow sublimates. This limits the total amount of blowing snow that can be sublimated and thus negatively feeds back on blowing snow sublimation. Following the method presented in the paper, forcing the blowing snow parameterization with an atmospheric model that ignore blowing snow and its sublimation neglects this negative feedback. In other words, this makes the atmosphere acting as an infinite sink for water vapor. Then,

even though the method presented relies on satellite observations, using raw moisture fields from such models to compute blowing snow sublimation very likely leads to significant overestimation. This appears to be a major limitation to the quantitative aspect of this work. Together with the arguments claimed in the discussion part, this certainly accounts for the large differences with previous model-derived estimates of Déry and Yau (2002) and Lenaerts et al. (2012). The overestimation of blowing snow sublimation compared to RACMO2 also seems questionable since the model has been shown to overestimate considerably the blowing snow flux and the resulting horizontal snow mass transport (see Lenaerts et al. 2014, their Figures 6b and 7c – pay attention to the difference between the left and right scales), suggesting an overestimation of the modelled sublimation.

Amory, C., Gallée, H., Naaim-Bouvet, F., Favier, V., Vignon, E., Picard, G., Trouvilliez, A., Piard, L., Genthon, C., and Bellot, H., 2017. Seasonal variations in drag coefficients over a sastrugi-covered snowfield in coastal East Antarctica. Boundary-Layer Meteorol., doi 10.1007/s10546-017-0242-5

Amory, C., Naaim-Bouvet, F., Gallée, H., and Vignon, E., 2016. Two well-marked cases of aerodynamic adjustment of sastrugi. The Cryosphere, 8, 1–8

Amory, C., Trouvilliez, A., Gallée, H., Favier, V., Naaim-Bouvet, F., Genthon, C., Agosta, C., Piard, L., and Bellot, H., 2015. Comparison between observed and simulated aeolian snow mass fluxes in Adélie Land, East Antarctica. The Cryosphere, 9, 1–12

Barral, H., Genthon, C., Trouvilliez, A., Brun, C., and Amory, C., 2014. Blowing snow in coastal Adélie Land, Antarctica: three atmospheric-moisture issues. The Cryosphere., 8, 1905–1919

Déry, S. J., and Yau, M. K., 2002. Large-scale mass balance effects of blowing snow and surface sublimation. J. Geophys. Res., 107(D23), 4679, 1–8. doi:10.1029/2001JD001251

Gallée, H., Trouvilliez, A., Agosta, C., Genthon, C., Favier, V., and Naaim-Bouvet, F., 2013. Transport of Snow by the Wind: A Comparison Between Observations in Adélie Land, Antarctica, and Simulations Made with the Regional Climate Model MAR. Boundary-Layer Meteorol., 146, 133–147

Grazioli, J., Genthon, C., Boudevillain, B., Duran-Alarcon, C., Del Guasta, M., Madeleine, J.-B., and Berne, A., 2017. Measurements of precipitation in Dumont d'Urville, Terre Adélie, East Antarctica. The Cryosphere Discussion. doi:10.5194/tc-2017-18

Lenaerts, J. T. M., van den Broeke, M. R., Déry, S. J., van Meijgaard, E., van de Berg, W. J., Palm, S. P., and Sanz Rodrigo, J., 2012. Modeling drifting snow in Antarctica with a regional climate model. 1. Methods and model evaluation. J. Geophys. Res., 117, D05108. doi:10.1029/2011JD016145

Lenaerts, J. T. M., Smeets, C. J. P. P, Nishimura, K., Eijkelboom, M., Boot, W., van den Broeke, M. R., and van de Berg, W. J., 2014. Drifting snow measurements on the Greenland Ice Sheet and their application for model evaluation. Cryosphere, 8(2), 801–814

Palerme, C., 2014. Etude des précipitations en Antarctique par téléléd étection radar, mesures in-situ, et intercomparaison de modèles de climat, 194 pp., Ph.D. thesis, Univ. Grenoble Alpes, Grenoble, France

Trouvilliez, A., Naaim-Bouvet, F., Genthon, C., Piard, L., Favier, V., Bellot, H., Agosta, C., Palerme, C., Amory, C., and Gallée, H., 2014. A novel experimental study of aeolian snow transport in Adélie Land (Antarctica). Cold Reg. Sci. Technol., 108, 125–138

Trouvilliez, A., Naaim-Bouvet, F., Bellot, H., Genthon, C., and Gallée, H., 2015. Evaluation of FlowCapt acoustic sensor for snowdrift measurements. J. Atmos. Ocean. Technol., 32(9), 1630-1641

---

## Referee Comment (RC2) · K. NISHIMURA (Referee) · 5 May 2017

This manuscript introduces the new approach to evaluate the blowing snow sublimation and the transport over Antarctica based on the CALIPSO satellite data and model re-analysis data product of MERRA-2. So far, authors contributed largely to reveal the extent and structures of the blowing snow over Antarctica with introducing the technique of satellite data analysis. It was certainly revolutionary in the blowing snow research community. Subjects are topics of conversation on these days and the idea described in this manuscript is also ambitious and attractive. Thus, the attitude should be highly evaluated. However, throughout the manuscript, numerous questions arose as shown below. These should be satisfactorily addressed before the paper can be accepted for

the publication. First of all, previous works are well reviewed in general except for the field observation lately carried out, such as Trouvilliez et al., 2014. Line 156: Here new logic to detect the blowing snow is introduced. I wonder how the new one gives effect on the blowing snow detections shown in the authors' previous publications (overestimate, misrecognize or negligible?). Line 185: Particle sizes of cirrus clouds are around 5 micron and are much less than the blowing snow particles. Is the same algorithm still applicable? Line 209: The definition of particle radius seems too rough, since the sublimation rates (vapor pressure) strongly rely on the particle radius. Although, latter part of the manuscript on line 450, authors estimated the error caused by the radius is 10 %, I believe the contribution cannot be assessed by such a simple product. Line 221: Observations by Mann et al. should be done much lower altitude than the one discussed here. Is it worth comparing? Fig. 2: Particle density profile shown here is very interesting, however, at the same time, it is very confusing. Please explain the physical mechanism why the particle density increases with height, as far as I know it should be opposite, and shows the maximum at 100 m. Line 239: Fk and Fd should be expressed in italic. Line 230: Kg → kg Line 256: Once the sublimation is generated in the specific grid, the properties of air, such as the temperature and the humidity, change and then flow into the next downstream grid. Obviously, amount of sublimation decreases along the flow except for the case of the drastic temperature rise. It looks this process is not taken into account in the procedure shown here. If this is the case, I am afraid it overestimates the sublimation amount largely and caused the difference with the previous research.. Further, the accuracy of the MERRA-2 data needs to be indicated because they are also the key for the following estimates. Since a number of AWSs exist over the Antarctica, comparisons with these measurements must be done, or at least be referred, and show the MERRA-2 data are precise enough as the input data. Line 347: Since the sublimated vapor does not always contribute on the sea level rise, I suppose this part is meaningless. Line 351: If this really the case, it is of great interest. Since the authors have all the ingredients (factors) to deduce the sublimation amount, the reason which brought the annual change can be conjectured, I believe. More detailed

considerations are recommended. Line 413: When we discuss the amount of sublimation and transport quantitatively, the blowing snow from the surface to 30 m, where this satellite technique is not applicable, cannot be neglected. As far as I know, the flux increases with decreasing height on the contrary to Fig. 2. Although the particles may suspend as high as 300 m, most of the transport concentrate within 0 to 30 m layer. It should be also taken into account in the error analysis. Recently, Huang et al.(2016) published the manuscript on Atmospheric Chemistry and Physics and discussed the impacts of moisture transport on drifting snow sublimation in the saltation layer. Thickness of the saltation layer is less than 0.1 m in usual, nevertheless, they say that the blowing (drifting) snow sublimation is important on the distribution and mass-energy balance of snow cover. Line 458: It the estimates of the error amounted to large as 40 %, the conclusion that 'the sublimation amount is about the twice the one of the previous studies' on line 478 is not always the case.
* * *

---

## Author Comment (AC1) · 28 Jun 2017

Reviewer 1, major comments I have strong reservations with regard to the use of simplified, steady-state parameterizations and meteorological forcing, which put a strong constraint on the resulting blowing snow sublimation. The authors use MERRA-2 temperature and RH data to derive sublimation rates, but these MERRA-2 data are (a) not at all evaluated over Antarctica, and – more importantly – (b) since MERRA-2 does not simulate blowing snow, do not imply the atmospheric effects of the well-documented self-limiting behavior of blowing snow sublimation, in which sublimation will lead to latent heat release to the atmosphere at the top of the blowing snow layer, in turn cooling

and moistening the atmosphere and limiting subsequent sublimation. This effect is expected to have a first-order negative effect on the sublimation (while retaining the blowing snow layer transport active, so this is not observable from space), and should – in some way or another – be included in this approach. I realize that the authors do not (and do not want to) utilize a model that includes this behavior, nor include blowing snow processes in MERRA-2. One option is to perform multiple sensitivity tests with gradually higher RHice values and lower temperatures, based on and in combination with a MERRA-2 near-surface climate evaluation at select stations over Antarctica. These sensitivity tests should be combined with varying other important parameters to determine sensitivity. For instance, why did the authors choose a fall velocity of 0.1 m/s? I would strongly suggest to expand the Section 4 and include a detaileddescription of the sensitivity tests.

The authors understand the reviewers concerns regarding the use of MERRA-2 reanalysis as the source of meteorological data for the blowing snow sublimation computations. Indeed, the MERRA-2 data have not been evaluated over Antarctica and the reviewers concerns are well justified. However, we have compared MERRA-2 relative humidity amounts with a few surface stations, dropsondes and other models and determined that MERRA-2 has, in general, a cold and moist bias. We have added section 2.1 to the paper which describes the MERRA-2 data and also show comparisons of MERRA-2 relative humidity and temperature to three surface stations and other models. The comparisons are shown in new figures 2 and 3.

The reviewer also notes that since MERRA-2 is a re-analysis based on the GEOS-5 model, it does not contain blowing snow physics that would capture processes such as the cooling and moistening of the blowing snow layer as sublimation proceeds. While we understand that blowing snow sublimation will act to moisten the layer, we also know that other processes operate, especially in deep blowing snow layers, that can act to reduce the humidity within the layer. For instance, entrainment of drier and warmer air at the top of the inversion and adiabatic warming associated with the descending

katabatic flow. Warming of the layer can also occur from trapping of longwave radiation. Most if not all of the observations showing how blowing snow will increase the moisture to or near saturation have been made at or below 10 m height. There are no observations through the depth of very deep blowing snow layers such as those presented here. Recently, we have been working with the Concordiasi dropsondes and have identified a number that fell through deep blowing snow layers. While this work is as yet too preliminary to present (or include in the pape), the observations suggest that typically the layer is not saturated.

Also noted by the reviewer, we do not necessarily want to use a model that contains blowing snow physics at this time. However, this may be an area that can be explored in the future. The Reviewer suggests a sensitivity study to see how increasing moisture will affect the blowing snow sublimation values. We regard this as a valuable suggestion and have added a new section 4.1 which addresses the effect of increasing moisture on blowing snow sublimation. We have also modified a paragraph in the conclusions section starting on line 541 of the revised paper. We state the major limitation of the study and admit the error in sublimation estimates could be as large as 40%.

As to why we chose a fall velocity of 0.1 m/s, it is a value close to the average fall velocity found by Mann et al., 2000. Granted these are for the surface layer and for larger particles. The particles in deep blowing snow layers will be smaller and have a lower fall speed. We did a test and found that reducing the fall speed by an order of magnitude (to 0.01 m/s) reduces the calculated sublimation by about 10%. Just between you and me, I wonder whether fall speed is the right thing to use here, considering that in many of these blowing snow episodes, wind speeds approach 20 m/s and wind speed and directional shear is very large. This has to produce incredible turbulence which will act to ventilate the particles well beyond simply falling at a constant speed through the atmosphere.

Reviewer 1 Minor comments: The writing should be improved in places and caution is warranted to very clearly describe the process the authors are referring to. Also, some

parts are clearly too speculative and should be revised (see below). L13: near-surface
Changed

L15: define surface mass balance Added Equation 1 on line 48 and text to explain on
lines 41-45

L17 and beyond: clearly mention the time period considered in this study Done on line
19

L23: blowing snow sublimation! Done

L29: 2006-2015 Done

L94: it would be helpful to mention all sublimation rates (also those from earlier liter-
ature)in the same units to facilitate comparison. Which time period are these from?
Changed all references to amount to mm swe yr-1. The Halley observations were form
1995 and 1996. This is added to text on line 94.

L108: 20,000 km – reference needed Added

L152: 1064/532 – include units Added

L189: (Walden et al., 2003) Changed

L233: How to go from blowing snow mixing ratio to extinction? What are the units of
this extinction, and why does they relate as alfa(z) = 3/2 qb(z)/r(z) ? We do not go from
blowing snow mixing ratio to extinction. Rather we first compute the extinction and from
that the particle number density N(z). From N(z) we get blowing snow mixing ratio via
equation 3. The units of extinction are 1/m

L240 and around: The use of MERRA-2 needs to be described here. How are T and
RH incorporated here? How is temporal and vertical interpolation dealt with? We have
added section 2.1 that describes the MERRA-2 data, its characteristics and how we
use it to obtain values at a given point in space and time. Note this begins on line 174
of revised text.

L241: Equations are not numbered Fixed

L321: It would be very helpful to plot the CloudSAT precipitation numbers and plot theratio sublimation/precipitation to guide this discussion. The Authors agree this would be a useful thing to do, but we do not have time to obtain the data and do the analysis.

L339: 419 Gt/yr – this is a different number than mentioned anywhere else. Should have been 393. This is now corrected.

L344-348: This is extremely speculative and contains the wrong translation from Gt/yrto mm sea level rise (360 Gt = 1 mm SLE). Most of the sublimation is probably recycled on the ice sheet, and of course Antarctic SMB is positive and dominated by precipitation. Please remove. Agreed. It has been removed.

L349-355: is the trend significant? Probably not, with significant inter-annual variability and only 10 years in the time series. If it is not significant, please remove. If it is significant, it would be useful to relate this to MERRA-2 T and RH averages. Removed

L357: clarify if you consider the grounded or total (include ice shelves) ice sheet. Clarified on line 379

L371: size of Texas – quantify. We have added the exact area of Texas to the text

Table I: Is the average from 2006-2015 (the full years)? Clarify. Table I now explains this in a footnote.

Figures 9 and 10 can be removed or moved to supplements. They do not contain any results that are necessary to be shown in a separate (main) figure. Agreed. We have moved them to the supplement.

Reviewer 2 comments: General comments This paper presents very interesting and unprecedented continent wide statistics of blowing snow over Antarctica from long-term satellite observations. These include estimations of blowing snow sublimation, a significant but poorly known component of the Antarctic surface mass balance. Such

works are essential for evaluation of atmospheric models from which the total surface mass budget of the ice sheet can be estimated. However, there are some important missing aspects and information in the study that I would like to report here. Of particular concern is the method from which sublimation estimates are computed. One possibly very significant source of error is an underestimation of atmospheric moisture by MERRA-2: the method does not take into account the fact that moisture from blowing snow sublimation is retained while air flows further through blowing snow regions, strongly reducing (or cutting if saturation is reached) any further sublimation of blowing snow downstream. MERRA-2 does not account for blowing snow sublimation, thus the method constantly resets air moisture to values for which blowing snow sublimation has never occurred, and very likely overestimates total sublimation.

We have added section 2.1 to the paper to show that MERRA-2 is moist compared to surface observations. Please see our comments to Reviewer 1 about the effect of blowing snow sublimation on the humidity of the layer. In short, there are no observations supporting the assertion that blowing snow sublimation will lead to saturation of the layer when you are dealing with layers 100-400 m thick. The only observations to support this are below 10 m height. Entrainment of dry and warm air from above, descending air in the katabatic flow and warming of the layer through absorption of longwave radiation are all process that can act to keep the layer from saturating.

Specific comments Observational studies on blowing snow in Antarctica are very scarce, to the extent that continuous measurements extending beyond a few weeks or months barely exist. However, considerable efforts have been made in the recent years on that specific topic, that you might have missed in your bibliography. An observation campaign dedicated to blowing snow has been run in January 2010 by the Laboratoire de Glaciologie et Géophysique de l'Environnement (LGGE, France) in coastal Adélie Land. Some of the collected data have been presented, for instance, in Trouvilliez et al. (2014), Barral et al. (2014) and Amory et al. (2016, 2017), and used for evaluation of preliminary modelling results (Gallée et al. 2013, Amory et al. 2015). Ground

measurements on the ice sheet have been performed using second-generation acoustic FlowCapt™ sensors. While these sensors have been shown to slightly underestimate the blowing snow flux compared to optical snow particle counters SPC-S7 in the French Alps, they remain excellent detectors of blowing snow occurrences (Trouvilliez et al. 2015). To date, up to 7 years (2010-2016) of continuous ground measurements of blowing snow frequency in coastal Adélie Land are available (for comparison with CALIPSO data). The dataset also includes (discontinuous) measurements of snow particle size performed since 2013 at 50-m height above the ground with a SPC at Dumont d'Urville station (see Palerme 2014). I'm part of the research team that has produced (and still does) these observations and I'm open to discuss it with the authors if they wish.

The Authors thank the Reviewer for this valuable information and references. We would like to explore these observations with the Reviewer in the near future.

P8, L222: Figure 2 shows an increase in particle density with height for the first 100 meters above the surface. This is surprising since the density of blowing snow particle is supposed to decrease as distance from the ground (i.e., from the particle source) increases (see for instance the strong decrease within the first 10 meters above the ground in Fig. 4 of Mann et al. 2000). Have you an idea of what can cause thisfeature? This is a very astute observation by the Reviewer. One of the things that one has to be concerned with when analyzing the CALIPSO data is contamination of the atmospheric measurement by the ground return signal. One has to be very careful to eliminate the ground return from the backscatter profile. If this is not done correctly, it will add (incorrectly) signal to the lowest bin of the profile (the bin directly above the ground). We have been very cautious about this and have probably erred on the conservative side. Thus some of the larger signals that may have been present in the bin directly above the ground may have been eliminated because of the possibility of ground signal contamination. But the calculated particle density profile in Fig 4 has a maximum 3 or 4 bins above the ground. We believe this is due to attenuation of the lidar signal as

it passes through the blowing snow layer. We have not attempted to correct for this, but if it were corrected it would tend to increase the particle number density as one approached the ground. We have added text to describe this and its effect.

P17, L431: In addition, clouds may be associated with precipitation which contributes moistening the dry surface air layer (Grazioli et al. 2017; http://www.the-cryospherediscuss. net/tc-2017-18/) and thus correspondingly reduces blowing snow sublimation. We cannot detect blowing snow occurring beneath precipitating clouds and such cases are not included here.

P17, L449 and onwards: Although this aspect is already partly discussed in the paper, estimating blowing snow sublimation by using MERRA-2 re-analysis fields of moisture could be misleading because i) re-analysed moisture near the surface could be underestimated and ii) no retro-action of sublimation on moisture is accounted for. Systematic dry biases in atmospheric models and meteorological (re-) analyses that do not account for blowing snow have been discussed in Barral et al. (2014). Using a 3-year dataset of ground measurements at a coastal location in Adélie Land, they showed (their Figure 6) for 3 modelling products that the moisture error in the near-surface layer for the continental grid point closest to the measurement location is much larger than 5% (as considered in the error analysis in Section 4), and that the 3 models fail to represent the observed increase of atmospheric moisture with wind speed. For instance, the moisture error almost averages 100% for the ECMWF operational analysis for wind speeds exceeding 12 m/s. It is likely that most meteorological and climate models ignoring blowing snow are affected by similar dry biases, at least over windy peripheral areas of East Antarctica where blowing snow is highly active. In addition, in the blowing snow layer the air quickly saturates as part of the blowing snow sublimates. This limits the total amount of blowing snow that can be sublimated and thus negatively feeds back on blowing snow sublimation. Following the method presented in the paper, forcing the blowing snow parameterization with an atmospheric model that ignore blowing snow and its sublimation neglects this negative feedback. In otherwords, this makes the atmosphere acting as an infinite sink for water vapor. Then, even though the method presented relies on satellite observations, using raw moisture fields from such models to compute blowing snow sublimation very likely leads to significant overestimation. This appears to be a major limitation to the quantitative aspect of this work. Together with the arguments claimed in the discussion part, this certainly accounts for the large differences with previous model-derived estimates of Déry and Yau (2002) and Lenaerts et al. (2012). The overestimation of blowing snow sublimation compared to RACMO2 also seems questionable since the model has been shown to overestimate considerably the blowing snow flux and the resulting horizontal snow mass transport (see Lenaerts et al. 2014, their Figures 6b and 7c – pay attention to the difference between the left and right scales), suggesting an overestimation of the modelled sublimation.

We have added section 2.1 to the paper which demonstrates that MERRA-2 is cold and moist with respect to surface observations and other models. Moreover, in ongoing research we have seen that this bias in MERRA-2 is not limited to the surface (2 m). Using the Concordiasi dropsondes, we have seen that MERRA-2 is cold and moist even at levels above the surface. Based on this we do not feel that MERRA-2 has a dry bias. Rather, it is likely too moist. Also please see the Author's comment to Reviewer 1 who had the same concern about not including blowing snow physics.

Reviewer 3 Comments: First of all, previous works are well reviewed in general except for the field observation lately carried out, such as Trouvilliez et al., 2014. This reference has been added to the text at line

Line 156: Here new logic to detect the blowing snow is introduced. I wonder how the new one gives effect on the blowing snow detections shown in the authors' previous publications (overestimate, misrecognize or negligible?). We have compared results with the previous algorithm and have seen very little change. This is stated on line 164 of the revised paper.

Line 185: Particle sizes of cirrus clouds are around 5 micron and are much less than the blowing snow particles. Is the same algorithm still applicable? The reviewer is mistaken. Cirrus cloud particle sizes range from 10 to 100 um and even much larger (see Heymefield et al., 2002 for instance)

Line 209: The definition of particle radius seems too rough, since the sublimation rates (vapor pressure) strongly rely on the particle radius. Although, latter part of the manuscript on line 450, authors estimated the error caused by the radius is 10 %, I believe the contribution cannot be assessed by such a simple product. The particle sizes produced by Equation 1 give sizes ranging from 40 (15 m) to 15 $\mu$m (500 m). Observations of blowing snow particle size made in the field are generally at 10 m or less. Until we get measurements that extend high up into the layer, this is about the best that can be done.

Line 221: Observations by Mann et al. should be done much lower altitude than the one discussed here. Is it worth comparing? Mann does have a plot that goes up to 11 m, and by extrapolation one can get an idea of the value at say 30 m. Also Mann has a theoretical plot of number density that goes up to 20m.

Fig. 2: Particle density profile shown here is very interesting, however, at the same time, it is very confusing. Please explain the physical mechanism why the particle density increases with height, as far as I know it should be opposite, and shows the maximum at 100 m. This is a good catch. I believe this is due to attenuation of the lidar return (the measured backscatter) as it traverses downward through the blowing snow layer. This is producing an underestimate of the extinction for the bins closest to the surface (say below 100 m). And the number density is computed from the extinction via equation 2. Note also that the radius of the particle is in the denominator of equation 2 and r decreases with height. This would cause N(z) to increase with height. The bottom line is I think you are right and the particle densities are lower than they should be in the lower 50-100 m. Note that this would lead to an under estimation of sublimation.

Line 239: Fk and Fd should be expressed in italic. I am not sure what you mean. In my document, they are in italic.

Line 230: Kg ! kg Fixed

Line 256: Once the sublimation is generated in the specific grid, the properties of air, such as the temperature and the humidity, change and then flow into the next downstream grid. Obviously, amount of sublimation decreases along the flow except for the case of the drastic temperature rise. It looks this process is not taken into account in the procedure shown here. If this is the case, I am afraid it overestimates the sublimation amount largely and caused the difference with the previous research.. Further, the accuracy of the MERRA-2 data needs to be indicated because they are also the key for the following estimates. Since a number of AWSs exist over the Antarctica, comparisons with these measurements must be done, or at least be referred, and show the MERRA-2 data are precise enough as the input data. This is a very good point and was also noted by other reviewers. There is really nothing we can do about the modification of the layer temperature and moisture by the blowing snow sublimation other than using a model that incorporates these processes. Even if we did so, there is no guarantee that when CALIPSO identifies a blowing snow layer, the model would know it is there and have correspondingly modified its temperature and moisture. Having said that, I believe this is an area in need of more research. Please note we have added sections 2.1 to describe the MERRA-2 data and compare with observations. We have also added section 4.1 which is a sensitivity study on the effect of increasing moisture on blowing snow sublimation.

Line 347: Since the sublimated vapor does not always contribute on the sea level rise, I suppose this part is meaningless. Agreed. This has been removed.

Line 351: If this really the case, it is of great interest. Since the authors have all the ingredients (factors) to deduce the sublimation amount, the reason which brought the annual change can be conjectured, I believe. More detailed considerations are

recommended. Another Reviewer suggested that this be removed as the time series is rather short

Line 413: When we discuss the amount of sublimation and transport quantitatively, the blowing snow from the surface to 30 m, where this satellite technique is not applicable, cannot be neglected. As far as I know, the flux increases with decreasing height on the contrary to Fig. 2. Although the particles may suspend as high as 300 m, most of the transport concentrate within 0 to 30 m layer. It should be also taken into account in the error analysis. Recently, Huang et al.(2016) published the manuscript on Atmospheric Chemistry and Physics and discussed the impacts of moisture transport on drifting snow sublimation in the saltation layer. Thickness of the saltation layer is less than 0.1 m in usual, nevertheless, they say that the blowing (drifting) snow sublimation is important on the distribution and mass-energy balance of snow cover. Thank you for this information. We have added text to the revised paper and cited Huang et al., 2016 on line 458 of revised text.

Line 458: It the estimates of the error amounted to large as 40 %, the conclusion that 'the sublimation amount is about the twice the one of the previous studies' on line 478 is not always the case. We have changed this wording

---

## Author Response (AR2)

Editor Decision: Reconsider after major revisions (11 Aug 2017) by Philip Marsh

Comments to the Author:

Dr. Palm. The second round of review comments have been returned to me, with both reviewers thanking you for the efforts made to improve the paper after the first round of reviews. However, both reviewers still have significant concerns that need to be satisfactorily addressed prior to publication. Although this is the second round of reviews, I am willing to allow another round of revisions if you wish. I would suggest that you address all of the reviewers ongoing concerns, including the following.

Reviewer #1:

The main concern is that you need to substantiate the claim that MERRA-2 is too moist/cold. One possible approach is to conduct a more thorough comparison of MERRA-2 RH with observations, and make sure consistent RH parameters are used in the different data sets. In addition, please consider the following as well.

We have added supplemental figures S1-S6 which show additional AWS comparisons with the MERRA-2 data. All data relative humidity data presented in these plots is with respect to ice.

Figure 2: what is the source of the AWS data? What is their name and elevation? Why are those 'randomly'?

The source of the AWS data is noted in the acknowledgement section of the paper. They were obtained from the University of Wisconsin site: ftp://amrc.ssec.wisc.edu/pub/aws/.

The elevation and site names are now on the  AWS-MERRA-2 comparison supplemental figures

Figure 3 a and 3b require significant improvement to address the following concerns. They basically show a big cloud of data and do not support any of the results. The authors should average the data in time and show e.g. monthly averages in comparison to stations. Also, what is the source of the station data? Are they quality controlled? Are they corrected for RH at low temperatures? Drawing conclusions from a comparison at PE (which is not a really typical 'Antarctic' station, since it is sheltered by topography, and blowing snow does not occur frequently) only does not provide evidence of any Antarctic-wide signal. If the authors want to justify this claim, it will require the inclusion of more weather stations.

We have taken the data shown in old figure 3 (Princess Elisabeth Station) and averaged it by month. This was then compared with the similarly averaged MERRA-2 data and the result is included in the revised paper as supplemental figure S7 and S8 and described in section 2.1 of the revised paper. The source of these data is noted in the acknowledgements. We have not as yet been able to determine how the data was quality controlled or corrected for low temperature. Other AWS sites have been included for comparison with MERRA-2

Also, even more importantly, does MERRA-2 show RH with respect to ice or to water? I suspect it's with respect to ice, and I know ERA-Interim shows RH with respect to water (even at freezing temperatures). This will largely explain the difference between the two reanalyses. Please check thoroughly and make sure they are consistent. If they are inconsistent, I would argue that the claim that MERRA-2 is too moist is no longer justifiable.

Note that old figure 3 has been removed from the revised paper and what used to be old figure 3b is now supplemental figure S9. Referring to that figure and your comment above, all 3 models are showing relative humidity with respect to water. MERRA-2 reports humidity with respect to water and when we use it in our computations of sublimation, we convert it to relative humidity with respect to ice.

Table I: please include uncertainties for each year.

Done

Line 499: How is 5% justified? The authors should carefully compare MERRA-2 to many AWS stations in Antarctica, and derive an uncertainty/error from that – see above.

We have added surface observation and MERRA-2 comparisons and determined that on average MERRA-2 relative humidity is 7% higher than observations.

Reviewer #2:

The major concern is that procedures to estimates the sublimation rates involve considerable errors and its evaluation is not sufficient to date.

Please address the following concerns:

Once the sublimation is generated in the specific grid, the properties of air, such as the temperature and the humidity, change and then flow into the next downstream grid. Obviously, amount of sublimation decreases along the flow except for the case of a drastic temperature rise. Please confirm whether this process is taken into account in the procedure described in the paper. If it is not, it seems likely that you overestimate sublimation. Could this be the cause of the difference with previous research.

With all due respect, what the reviewer is asking us to do is beyond the scope or intent of this paper. We are not employing a model that has blowing snow physics or trying to model how the air is modified from the sublimation. We understand that sublimation occurring within a closed parcel of air will eventually saturate the layer. However, nature does not close its parcels of air in an air tight box. There are other processes that occur and for which models do not have a good handle on. These processes will act to keep the layer from reaching saturation. Processes such as entrainment of dry and warmer air from above the layer, adiabatic warming in a katabatic flow, and warming of the layer due to trapping of longwave radiation. Keep in mind, that the only observations that show relative humidity reaching saturation during blowing snow events are made at the surface. This paper deals with blowing snow layers on average 120 m thick, and often much higher than that. How do these layers grow? They grow by entraining the air from above which is almost always warmer and dryer.

There is a concern that the procedure to evaluate the total error is no relevant. Probably more careful analysis is needed. At a minimum, estimated errors of +/-50 % should be declared in the abstract as well.

We respectfully disagree with the reviewer's assessment that our analysis of error is not relevant. We have carefully looked at error contributions from every source. We feel we have done a thorough job with that. The 50% error is stated in the abstract.

Thank you ,

Philip Marsh

Line 212: there is a need to better explain why the same algorithm for cirrus clouds is applicable for the larger blowing snow particles.

As we explained in our previous response to the reviewer, cirrus cloud particles are in the same size range or larger than blowing snow particles. Blowing snow particles, especially above 30 m are generally smaller than cirrus cloud particles. Cirrus cloud particle sizes range from 10 to 100 um and even much larger (see Heymsfield et al., 2002 for instance)

[revised manuscript text omitted]